# A Convergence Analysis of Gradient Descent for Deep Linear Neural Networks

**Sanjeev Arora**
Princeton University and Institute for Advanced Study
`arora@cs.princeton.edu`

**Nadav Cohen**
Institute for Advanced Study
`cohennadav@ias.edu`

**Noah Golowich**
Harvard University
`ngolowich@college.harvard.edu`

**Wei Hu**
Princeton University
`huwei@cs.princeton.edu`

## Abstract

We analyze speed of convergence to global optimum for gradient descent training a deep linear neural network (parameterized as $x \mapsto W_N W_{N-1} \cdots W_1 x$) by minimizing the $\ell_2$ loss over whitened data. Convergence at a linear rate is guaranteed when the following hold: *(i)* dimensions of hidden layers are at least the minimum of the input and output dimensions; *(ii)* weight matrices at initialization are approximately balanced; and *(iii)* the initial loss is smaller than the loss of any rank-deficient solution. The assumptions on initialization (conditions *(ii)* and *(iii)*) are necessary, in the sense that violating any one of them may lead to convergence failure. Moreover, in the important case of output dimension 1, *i.e.* scalar regression, they are met, and thus convergence to global optimum holds, with constant probability under a random initialization scheme. Our results significantly extend previous analyses, *e.g.*, of deep linear residual networks (Bartlett et al., 2018).

## 1 Introduction

Deep learning builds upon the mysterious ability of gradient-based optimization methods to solve related non-convex problems. Immense efforts are underway to mathematically analyze this phenomenon. The prominent *landscape* approach focuses on special properties of critical points (*i.e.* points where the gradient of the objective function vanishes) that will imply convergence to global optimum. Several papers (*e.g.* Ge et al. (2015); Lee et al. (2016)) have shown that (given certain smoothness properties) it suffices for critical points to meet the following two conditions: *(i) no poor local minima* — every local minimum is close in its objective value to a global minimum; and *(ii) strict saddle property* — every critical point that is not a local minimum has at least one negative eigenvalue to its Hessian. While condition *(i)* does not always hold (*cf.* Safran and Shamir (2018)), it has been established for various simple settings (*e.g.* Soudry and Carmon (2016); Kawaguchi (2016)). Condition *(ii)* on the other hand seems less plausible, and is in fact provably false for models with three or more layers (*cf.* Kawaguchi (2016)), *i.e.* for *deep* networks. It has only been established for problems involving *shallow* (two layer) models, *e.g.* matrix factorization (Ge et al. (2016); Du et al. (2018a)). The landscape approach as currently construed thus suffers from inherent limitations in proving convergence to global minimum for deep networks.

A potential path to circumvent this obstacle lies in realizing that landscape properties matter only in the vicinity of trajectories that can be taken by the optimizer, which may be a negligible portion of the overall parameter space. Several papers (*e.g.* Saxe et al. (2014); Arora et al. (2018)) have taken this *trajectory-based* approach, primarily in the context of *linear neural networks* — fully-connected neural networks with linear activation. Linear networks are trivial from a representational perspective, but not so in terms of optimization — they lead to non-convex training problems with multiple minima and saddle points. Through a mix of theory and experiments, Arora et al. (2018) argued that such non-convexities may in fact be beneficial for gradient descent, in the sense that sometimes, adding (redundant) linear layers to a classic linear prediction model can accelerate the optimization. This phenomenon challenges the holistic landscape view, by which convex problems are always preferable to non-convex ones.

Even in the linear network setting, a rigorous proof of efficient convergence to global minimum has proved elusive. One recent progress is the analysis of Bartlett et al. (2018) for *linear residual networks* — a particular subclass of linear neural networks in which the input, output and all hidden dimensions are equal, and all layers are initialized to be the identity matrix (*cf.* Hardt and Ma (2016)). Through a trajectory-based analysis of gradient descent minimizing $\ell_2$ loss over a whitened dataset (see Section 2), Bartlett et al. (2018) show that convergence to global minimum at a *linear rate* — loss is less than $\epsilon > 0$ after $\mathcal{O}(\log \frac{1}{\epsilon})$ iterations — takes place if one of the following holds: *(i)* the objective value at initialization is sufficiently close to a global minimum; or *(ii)* a global minimum is attained when the product of all layers is positive definite.

The current paper carries out a trajectory-based analysis of gradient descent for general deep linear neural networks, covering the residual setting of Bartlett et al. (2018), as well as many more settings that better match practical deep learning. Our analysis draws upon the trajectory characterization of Arora et al. (2018) for gradient flow (infinitesimally small learning rate), together with significant new ideas necessitated due to discrete updates. Ultimately, we show that when minimizing $\ell_2$ loss of a deep linear network over a whitened dataset, gradient descent converges to the global minimum, at a linear rate, provided that the following conditions hold: *(i)* the dimensions of hidden layers are greater than or equal to the minimum between those of the input and output; *(ii)* layers are initialized to be *approximately balanced* (see Definition 1) — this is met under commonplace near-zero, as well as residual (identity) initializations; and *(iii)* the initial loss is smaller than any loss obtainable with rank deficiencies — this condition will hold with probability close to $0.5$ if the output dimension is $1$ (scalar regression) and standard (random) near-zero initialization is employed. Our result applies to networks with arbitrary depth and input/output dimensions, as well as any configuration of hidden layer widths that does not force rank deficiency (*i.e.* that meets condition *(i)*). The assumptions on initialization (conditions *(ii)* and *(iii)*) are necessary, in the sense that violating any one of them may lead to convergence failure. Moreover, in the case of scalar regression, they are met with constant probability under a random initialization scheme. We are not aware of any similarly general analysis for efficient convergence of gradient descent to global minimum in deep learning.

The remainder of the paper is organized as follows. In Section 2 we present the problem of gradient descent training a deep linear neural network by minimizing the $\ell_2$ loss over a whitened dataset. Section 3 formally states our assumptions, and presents our convergence analysis. Key ideas brought forth by our analysis are demonstrated empirically in Section 4. Section 5 gives a review of relevant literature, including a detailed comparison of our results against those of Bartlett et al. (2018). Finally, Section 6 concludes.

## 2 GRADIENT DESCENT FOR DEEP LINEAR NEURAL NETWORKS

We denote by $\|\mathbf{v}\|$ the Euclidean norm of a vector $\mathbf{v}$, and by $\|A\|_F$ the Frobenius norm of a matrix $A$.

We are given a training set $\{(\mathbf{x}^{(i)}, \mathbf{y}^{(i)})\}_{i=1}^m \subset \mathbb{R}^{d_x} \times \mathbb{R}^{d_y}$, and would like to learn a hypothesis (predictor) from a parametric family $\mathcal{H} := \{h_\theta : \mathbb{R}^{d_x} \to \mathbb{R}^{d_y} \,|\, \theta \in \Theta\}$ by minimizing the $\ell_2$ loss:[1]

$$\min_{\theta \in \Theta} \; L(\theta) := \frac{1}{2m} \sum\nolimits_{i=1}^m \|h_\theta(\mathbf{x}^{(i)}) - \mathbf{y}^{(i)}\|^2 \,.$$

When the parametric family in question is the class of linear predictors, *i.e.* $\mathcal{H} = \{\mathbf{x} \mapsto W\mathbf{x} \,|\, W \in \mathbb{R}^{d_y \times d_x}\}$, the training loss may be written as $L(W) = \frac{1}{2m}\|WX - Y\|_F^2$, where $X \in \mathbb{R}^{d_x \times m}$ and $Y \in \mathbb{R}^{d_y \times m}$ are matrices whose columns hold instances and labels respectively. Suppose now that the dataset is *whitened*, *i.e.* has been transformed such that the empirical (uncentered) covariance matrix for instances — $\Lambda_{xx} := \frac{1}{m} X X^\top \in \mathbb{R}^{d_x \times d_x}$ — is equal to identity. Standard calculations (see Appendix A) show that in this case:

$$L(W) = \frac{1}{2} \|W - \Lambda_{yx}\|_F^2 + c \,, \tag{1}$$

---

[1]Much of the analysis in this paper can be extended to loss types other than $\ell_2$. In particular, the notion of deficiency margin (Definition 2) can be generalized to account for any convex loss, and, so long as the loss is differentiable, a convergence result analogous to Theorem 1 will hold in the idealized setting of perfect initial balancedness and infinitesimally small learning rate (see proof of Lemma 1). We leave to future work treatment of approximate balancedness and discrete updates in this general setting.

where $\Lambda_{yx} := \frac{1}{m} Y X^\top \in \mathbb{R}^{d_y \times d_x}$ is the empirical (uncentered) cross-covariance matrix between instances and labels, and $c$ is a constant (that does not depend on $W$). Denoting $\Phi := \Lambda_{yx}$ for brevity, we have that for linear models, minimizing $\ell_2$ loss over whitened data is equivalent to minimizing the squared Frobenius distance from a *target matrix* $\Phi$:

$$\min_{W \in \mathbb{R}^{d_y \times d_x}} \ L^1(W) := \frac{1}{2} \left\| W - \Phi \right\|_F^2 . \tag{2}$$

Our interest in this work lies on *linear neural networks* — fully-connected neural networks with linear activation. A depth-$N$ ($N \in \mathbb{N}$) linear neural network with hidden widths $d_1, \ldots, d_{N-1} \in \mathbb{N}$ corresponds to the parametric family of hypotheses $\mathcal{H} := \{ \mathbf{x} \mapsto W_N W_{N-1} \cdots W_1 \mathbf{x} \,|\, W_j \in \mathbb{R}^{d_j \times d_{j-1}}, \, j = 1, \ldots, N \}$, where $d_0 := d_x$, $d_N := d_y$. Similarly to the case of a (directly parameterized) linear predictor (Equation (2)), with a linear neural network, minimizing $\ell_2$ loss over whitened data can be cast as squared Frobenius approximation of a target matrix $\Phi$:

$$\min_{W_j \in \mathbb{R}^{d_j \times d_{j-1}}, \, j=1,\ldots,N} \ L^N(W_1, \ldots, W_N) := \frac{1}{2} \left\| W_N W_{N-1} \cdots W_1 - \Phi \right\|_F^2 . \tag{3}$$

Note that the notation $L^N(\cdot)$ is consistent with that of Equation (2), as a network with depth $N = 1$ precisely reduces to a (directly parameterized) linear model.

We focus on studying the process of training a deep linear neural network by *gradient descent*, *i.e.* of tackling the optimization problem in Equation (3) by iteratively applying the following updates:

$$W_j(t+1) \leftarrow W_j(t) - \eta \frac{\partial L^N}{\partial W_j}\big(W_1(t), \ldots, W_N(t)\big) \quad , j = 1, \ldots, N \quad , t = 0, 1, 2, \ldots \ , \tag{4}$$

where $\eta > 0$ is a configurable learning rate. In the case of depth $N = 1$, the training problem in Equation (3) is smooth and strongly convex, thus it is known (*cf.* Boyd and Vandenberghe (2004)) that with proper choice of $\eta$, gradient descent converges to global minimum at a linear rate. In contrast, for any depth greater than 1, Equation (3) comprises a fundamentally non-convex program, and the convergence properties of gradient descent are highly non-trivial. Apart from the case $N = 2$ (shallow network), one cannot hope to prove convergence via landscape arguments, as the strict saddle property is provably violated (see Section 1). We will see in Section 3 that a direct analysis of the trajectories taken by gradient descent can succeed in this arena, providing a guarantee for linear rate convergence to global minimum.

We close this section by introducing additional notation that will be used in our analysis. For an arbitrary matrix $A$, we denote by $\sigma_{max}(A)$ and $\sigma_{min}(A)$ its largest and smallest (respectively) singular values.[2] For $d \in \mathbb{N}$, we use $I_d$ to signify the identity matrix in $\mathbb{R}^{d \times d}$. Given weights $W_1, \ldots, W_N$ of a linear neural network, we let $W_{1:N}$ be the direct parameterization of the *end-to-end* linear mapping realized by the network, *i.e.* $W_{1:N} := W_N W_{N-1} \cdots W_1$. Note that $L^N(W_1, \ldots, W_N) = L^1(W_{1:N})$, meaning the loss associated with a depth-$N$ network is equal to the loss of the corresponding end-to-end linear model. In the context of gradient descent, we will oftentimes use $\ell(t)$ as shorthand for the loss at iteration $t$:

$$\ell(t) := L^N(W_1(t), \ldots, W_N(t)) = L^1(W_{1:N}(t)) . \tag{5}$$

## 3 CONVERGENCE ANALYSIS

In this section we establish convergence of gradient descent for deep linear neural networks (Equations (4) and (3)) by directly analyzing the trajectories taken by the algorithm. We begin in Subsection 3.1 with a presentation of two concepts central to our analysis: *approximate balancedness* and *deficiency margin*. These facilitate our main convergence theorem, delivered in Subsection 3.2. We conclude in Subsection 3.3 by deriving a convergence guarantee that holds with constant probability over a random initialization.

### 3.1 APPROXIMATE BALANCEDNESS AND DEFICIENCY MARGIN

In our context, the notion of approximate balancedness is formally defined as follows:

---

[2]If $A \in \mathbb{R}^{d \times d'}$, $\sigma_{min}(A)$ stands for the $\min\{d, d'\}$-th largest singular value. Recall that singular values are always non-negative.

**Definition 1.** *For $\delta \geq 0$, we say that the matrices $W_j \in \mathbb{R}^{d_j \times d_{j-1}}$, $j{=}1,\ldots,N$, are $\delta$-*balanced *if:*

$$\left\| W_{j+1}^\top W_{j+1} - W_j W_j^\top \right\|_F \leq \delta \quad , \forall j \in \{1,\ldots,N-1\}.$$

Note that in the case of 0-balancedness, *i.e.* $W_{j+1}^\top W_{j+1} = W_j W_j^\top$, $\forall j \in \{1,\ldots,N-1\}$, all matrices $W_j$ share the same set of non-zero singular values. Moreover, as shown in the proof of Theorem 1 in Arora et al. (2018), this set is obtained by taking the $N$-th root of each non-zero singular value in the end-to-end matrix $W_{1:N}$. We will establish approximate versions of these facts for $\delta$-balancedness with $\delta > 0$, and admit their usage by showing that if the weights of a linear neural network are initialized to be approximately balanced, they will remain that way throughout the iterations of gradient descent. The condition of approximate balancedness at initialization is trivially met in the special case of linear residual networks ($d_0 = \cdots = d_N = d$ and $W_1(0) = \cdots = W_N(0) = I_d$). Moreover, as Claim 2 in Appendix B shows, for a given $\delta > 0$, the customary initialization via random Gaussian distribution with mean zero leads to approximate balancedness with high probability if the standard deviation is sufficiently small.

The second concept we introduce — deficiency margin — refers to how far a ball around the target is from containing rank-deficient (*i.e.* low rank) matrices.

**Definition 2.** *Given a target matrix $\Phi \in \mathbb{R}^{d_N \times d_0}$ and a constant $c > 0$, we say that a matrix $W \in \mathbb{R}^{d_N \times d_0}$ has* deficiency margin $c$ *with respect to* $\Phi$ *if:*[3]

$$\|W - \Phi\|_F \leq \sigma_{min}(\Phi) - c. \tag{6}$$

The term "deficiency margin" alludes to the fact that if Equation (6) holds, every matrix $W'$ whose distance from $\Phi$ is no greater than that of $W$, has singular values $c$-bounded away from zero:

**Claim 1.** *Suppose $W$ has deficiency margin $c$ with respect to $\Phi$. Then, any matrix $W'$ (of same size as $\Phi$ and $W$) for which $\|W' - \Phi\|_F \leq \|W - \Phi\|_F$ satisfies $\sigma_{min}(W') \geq c$.*

*Proof.* Our proof relies on the inequality $\sigma_{min}(A+B) \geq \sigma_{min}(A) - \sigma_{max}(B)$ — see Appendix D.1. $\square$

We will show that if the weights $W_1,\ldots,W_N$ are initialized such that (they are approximately balanced and) the end-to-end matrix $W_{1:N}$ has deficiency margin $c > 0$ with respect to the target $\Phi$, convergence of gradient descent to global minimum is guaranteed.[4] Moreover, the convergence will outpace a particular rate that gets faster when $c$ grows larger. This suggests that from a theoretical perspective, it is advantageous to initialize a linear neural network such that the end-to-end matrix has a large deficiency margin with respect to the target. Claim 3 in Appendix B provides information on how likely deficiency margins are in the case of a single output model (scalar regression) subject to customary zero-centered Gaussian initialization. It shows in particular that if the standard deviation of the initialization is sufficiently small, the probability of a deficiency margin being met is close to $0.5$; on the other hand, for this deficiency margin to have considerable magnitude, a non-negligible standard deviation is required.

Taking into account the need for both approximate balancedness and deficiency margin at initialization, we observe a delicate trade-off under the common setting of Gaussian perturbations around zero: if the standard deviation is small, it is likely that weights be highly balanced and a deficiency margin be met; however overly small standard deviation will render high magnitude for the deficiency margin improbable, and therefore fast convergence is less likely to happen; on the opposite end, large standard deviation jeopardizes both balancedness and deficiency margin, putting the entire convergence at risk. This trade-off is reminiscent of empirical phenomena in deep learning, by

---

[3]Note that deficiency margin $c > 0$ with respect to $\Phi$ implies $\sigma_{min}(\Phi) > 0$, *i.e.* $\Phi$ has full rank. Our analysis can be extended to account for rank-deficient $\Phi$ by replacing $\sigma_{min}(\Phi)$ in Equation (6) with the smallest positive singular value of $\Phi$, and by requiring that the end-to-end matrix $W_{1:N}$ be initialized such that its left and right null spaces coincide with those of $\Phi$. Relaxation of this requirement is a direction for future work.

[4]In fact, a deficiency margin implies that all critical points in the respective sublevel set (set of points with smaller loss value) are global minima. This however is far from sufficient for proving convergence, as sublevel sets are unbounded, and the loss landscape over them is non-convex and non-smooth. Indeed, we show in Appendix C that deficiency margin alone is not enough to ensure convergence — without approximate balancedness, the lack of smoothness can cause divergence.

which small initialization can bring forth efficient convergence, while if exceedingly small, rate of convergence may plummet ("vanishing gradient problem"), and if made large, divergence becomes inevitable ("exploding gradient problem"). The common resolution of residual connections (He et al., 2016) is analogous in our context to linear residual networks, which ensure perfect balancedness, and allow large deficiency margin if the target is not too far from identity.

## 3.2 MAIN THEOREM

Using approximate balancedness (Definition 1) and deficiency margin (Definition 2), we present our main theorem — a guarantee for linear convergence to global minimum:

**Theorem 1.** *Assume that gradient descent is initialized such that the end-to-end matrix $W_{1:N}(0)$ has deficiency margin $c > 0$ with respect to the target $\Phi$, and the weights $W_1(0), \ldots, W_N(0)$ are $\delta$-balanced with $\delta = c^2 / \left(256 \cdot N^3 \cdot \|\Phi\|_F^{2(N-1)/N}\right)$. Suppose also that the learning rate $\eta$ meets:*

$$\eta \leq \frac{c^{(4N-2)/N}}{6144 \cdot N^3 \cdot \|\Phi\|_F^{(6N-4)/N}} . \tag{7}$$

*Then, for any $\epsilon > 0$ and:*

$$T \geq \frac{1}{\eta \cdot c^{2(N-1)/N}} \cdot \log\left(\frac{\ell(0)}{\epsilon}\right) , \tag{8}$$

*the loss at iteration $T$ of gradient descent — $\ell(T)$ — is no greater than $\epsilon$.*

### 3.2.1 ON THE ASSUMPTIONS MADE

The assumptions made in Theorem 1 — approximate balancedness and deficiency margin at initialization — are both necessary, in the sense that violating any one of them may lead to convergence failure. We demonstrate this in Appendix C. In the special case of linear residual networks (uniform dimensions and identity initialization), a sufficient condition for the assumptions to be met is that the target matrix have (Frobenius) distance less than $0.5$ from identity. This strengthens one of the central results in Bartlett et al. (2018) (see Section 5). For a setting of random near-zero initialization, we present in Subsection 3.3 a scheme that, when the output dimension is $1$ (scalar regression), ensures assumptions are satisfied (and therefore gradient descent efficiently converges to global minimum) with constant probability. It is an open problem to fully analyze gradient descent under the common initialization scheme of zero-centered Gaussian perturbations applied to each layer independently. We treat this scenario in Appendix B, providing quantitative results concerning the likelihood of each assumption (approximate balancedness or deficiency margin) being met individually. However the question of how likely it is that both assumptions be met simultaneously, and how that depends on the standard deviation of the Gaussian, is left for future work.

An additional point to make is that Theorem 1 poses a structural limitation on the linear neural network. Namely, it requires the dimension of each hidden layer ($d_i$, $i = 1, \ldots, N-1$) to be greater than or equal to the minimum between those of the input ($d_0$) and output ($d_N$). Indeed, in order for the initial end-to-end matrix $W_{1:N}(0)$ to have deficiency margin $c > 0$, it must (by Claim 1) have full rank, and this is only possible if there is no intermediate dimension $d_i$ smaller than $\min\{d_0, d_N\}$. We make no other assumptions on network architecture (depth, input/output/hidden dimensions).

### 3.2.2 PROOF

The cornerstone upon which Theorem 1 rests is the following lemma, showing non-trivial descent whenever $\sigma_{min}(W_{1:N})$ is bounded away from zero:

**Lemma 1.** *Under the conditions of Theorem 1, we have that for every $t = 0, 1, 2, \ldots$:[5]*

$$\ell(t+1) \leq \ell(t) - \frac{\eta}{2} \cdot \sigma_{min}\big(W_{1:N}(t)\big)^{\frac{2(N-1)}{N}} \cdot \left\|\frac{dL^1}{dW}\big(W_{1:N}(t)\big)\right\|_F^2 . \tag{9}$$

---

[5]Note that the term $\frac{dL^1}{dW}(W_{1:N}(t))$ below stands for the gradient of $L^1(\cdot)$ — a convex loss over (directly parameterized) linear models (Equation (2)) — at the point $W_{1:N}(t)$ — the end-to-end matrix of the network at iteration $t$. It is therefore (see Equation (5)) non-zero anywhere but at a global minimum.

*Proof of Lemma 1 (in idealized setting; for complete proof see Appendix D.2).* We prove the lemma here for the idealized setting of perfect initial balancedness ($\delta = 0$):

$$W_{j+1}^\top(0)W_{j+1}(0) = W_j(0)W_j^\top(0) \quad , \forall j \in \{1, \ldots, N-1\} \, ,$$

and infinitesimally small learning rate ($\eta \to 0^+$) — *gradient flow*:

$$\dot{W}_j(\tau) = -\frac{\partial L^N}{\partial W_j}\big(W_1(\tau), \ldots, W_N(\tau)\big) \quad , j = 1, \ldots, N \quad , \tau \in [0, \infty) \, ,$$

where $\tau$ is a continuous time index, and dot symbol (in $\dot{W}_j(\tau)$) signifies derivative with respect to time. The complete proof, for the realistic case of approximate balancedness and discrete updates ($\delta, \eta > 0$), is similar but much more involved, and appears in Appendix D.2.

Recall that $\ell(t)$ — the objective value at iteration $t$ of gradient descent — is equal to $L^1(W_{1:N}(t))$ (see Equation (5)). Accordingly, for the idealized setting in consideration, we would like to show:

$$\frac{d}{d\tau}L^1\big(W_{1:N}(\tau)\big) \leq -\frac{1}{2}\sigma_{min}\big(W_{1:N}(\tau)\big)^{\frac{2(N-1)}{N}} \cdot \left\|\frac{dL^1}{dW}\big(W_{1:N}(\tau)\big)\right\|_F^2 . \tag{10}$$

We will see that a stronger version of Equation (10) holds, namely, one without the $1/2$ factor (which only appears due to discretization).

By (Theorem 1 and Claim 1 in) Arora et al. (2018), the weights $W_1(\tau), \ldots, W_N(\tau)$ remain balanced throughout the entire optimization, and that implies the end-to-end matrix $W_{1:N}(\tau)$ moves according to the following differential equation:

$$vec\big(\dot{W}_{1:N}(\tau)\big) = -P_{W_{1:N}(\tau)} \cdot vec\left(\frac{dL^1}{dW}\big(W_{1:N}(\tau)\big)\right) , \tag{11}$$

where $vec(A)$, for an arbitrary matrix $A$, stands for vectorization in column-first order, and $P_{W_{1:N}(\tau)}$ is a positive semidefinite matrix whose eigenvalues are all greater than or equal to $\sigma_{min}(W_{1:N}(\tau))^{2(N-1)/N}$. Taking the derivative of $L^1(W_{1:N}(\tau))$ with respect to time, we obtain the sought-after Equation (10) (with no $1/2$ factor):

$$
\begin{aligned}
\frac{d}{d\tau}L^1\big(W_{1:N}(\tau)\big) &= \left\langle vec\left(\frac{dL^1}{dW}\big(W_{1:N}(\tau)\big)\right), vec\big(\dot{W}_{1:N}(\tau)\big)\right\rangle \\
&= \left\langle vec\left(\frac{dL^1}{dW}\big(W_{1:N}(\tau)\big)\right), -P_{W_{1:N}(\tau)} \cdot vec\left(\frac{dL^1}{dW}\big(W_{1:N}(\tau)\big)\right)\right\rangle \\
&\leq -\sigma_{min}\big(W_{1:N}(\tau)\big)^{\frac{2(N-1)}{N}} \cdot \left\|vec\left(\frac{dL^1}{dW}\big(W_{1:N}(\tau)\big)\right)\right\|^2 \\
&= -\sigma_{min}\big(W_{1:N}(\tau)\big)^{\frac{2(N-1)}{N}} \cdot \left\|\frac{dL^1}{dW}\big(W_{1:N}(\tau)\big)\right\|_F^2 .
\end{aligned}
$$

The first transition here (equality) is an application of the chain rule; the second (equality) plugs in Equation (11); the third (inequality) results from the fact that the eigenvalues of the symmetric matrix $P_{W_{1:N}(\tau)}$ are no smaller than $\sigma_{min}(W_{1:N}(\tau))^{2(N-1)/N}$ (recall that $\|\cdot\|$ stands for Euclidean norm); and the last (equality) is trivial — $\|A\|_F = \|vec(A)\|$ for any matrix $A$. $\qquad\square$

With Lemma 1 established, the proof of Theorem 1 readily follows:

*Proof of Theorem 1.* By the definition of $L^1(\cdot)$ (Equation (2)), for any $W \in \mathbb{R}^{d_N \times d_0}$:

$$\frac{dL^1}{dW}(W) = W - \Phi \quad \Longrightarrow \quad \left\|\frac{dL^1}{dW}(W)\right\|_F^2 = 2 \cdot L^1(W) .$$

Plugging this into Equation (9) while recalling that $\ell(t) = L^1(W_{1:N}(t))$ (Equation (5)), we have (by Lemma 1) that for every $t = 0, 1, 2, \ldots$ :

$$L^1\big(W_{1:N}(t+1)\big) \leq L^1\big(W_{1:N}(t)\big) \cdot \left(1 - \eta \cdot \sigma_{min}\big(W_{1:N}(t)\big)^{\frac{2(N-1)}{N}}\right) .$$

Since the coefficients $1 - \eta \cdot \sigma_{min}(W_{1:N}(t))^{\frac{2(N-1)}{N}}$ are necessarily non-negative (otherwise would contradict non-negativity of $L^1(\cdot)$), we may unroll the inequalities, obtaining:

$$L^1\big(W_{1:N}(t+1)\big) \leq L^1\big(W_{1:N}(0)\big) \cdot \prod_{t'=0}^{t} \left(1 - \eta \cdot \sigma_{min}\big(W_{1:N}(t')\big)^{\frac{2(N-1)}{N}}\right). \qquad (12)$$

Now, this in particular means that for every $t' = 0, 1, 2, \ldots$:

$$L^1\big(W_{1:N}(t')\big) \leq L^1\big(W_{1:N}(0)\big) \implies \|W_{1:N}(t') - \Phi\|_F \leq \|W_{1:N}(0) - \Phi\|_F .$$

Deficiency margin $c$ of $W_{1:N}(0)$ along with Claim 1 thus imply $\sigma_{min}\big(W_{1:N}(t')\big) \geq c$, which when inserted back into Equation (12) yields, for every $t = 1, 2, 3, \ldots$:

$$L^1\big(W_{1:N}(t)\big) \leq L^1\big(W_{1:N}(0)\big) \cdot \left(1 - \eta \cdot c^{\frac{2(N-1)}{N}}\right)^t . \qquad (13)$$

$\eta \cdot c^{\frac{2(N-1)}{N}}$ is obviously non-negative, and it is also no greater than 1 (otherwise would contradict non-negativity of $L^1(\cdot)$). We may therefore incorporate the inequality $1 - \eta \cdot c^{2(N-1)/N} \leq \exp\big(-\eta \cdot c^{2(N-1)/N}\big)$ into Equation (13):

$$L^1\big(W_{1:N}(t)\big) \leq L^1\big(W_{1:N}(0)\big) \cdot \exp\big(-\eta \cdot c^{2(N-1)/N} \cdot t\big),$$

from which it follows that $L^1(W_{1:N}(t)) \leq \epsilon$ if:

$$t \geq \frac{1}{\eta \cdot c^{2(N-1)/N}} \cdot \log\left(\frac{L^1(W_{1:N}(0))}{\epsilon}\right) .$$

Recalling again that $\ell(t) = L^1(W_{1:N}(t))$ (Equation (5)), we conclude the proof. $\qquad \square$

### 3.3 BALANCED INITIALIZATION

We define the following procedure, *balanced initialization*, which assigns weights randomly while ensuring perfect balancedness:

**Procedure 1** (Balanced initialization). *Given $d_0, d_1, \ldots, d_N \in \mathbb{N}$ such that $\min\{d_1, \ldots, d_{N-1}\} \geq \min\{d_0, d_N\}$ and a distribution $\mathcal{D}$ over $d_N \times d_0$ matrices, a* balanced initialization *of $W_j \in \mathbb{R}^{d_j \times d_{j-1}}$, $j=1, \ldots, N$, assigns these weights as follows:*

*(i) Sample $A \in \mathbb{R}^{d_N \times d_0}$ according to $\mathcal{D}$.*

*(ii) Take singular value decomposition $A = U\Sigma V^\top$, where $U \in \mathbb{R}^{d_N \times \min\{d_0, d_N\}}$, $V \in \mathbb{R}^{d_0 \times \min\{d_0, d_N\}}$ have orthonormal columns, and $\Sigma \in \mathbb{R}^{\min\{d_0, d_N\} \times \min\{d_0, d_N\}}$ is diagonal and holds the singular values of $A$.*

*(iii) Set $W_N \simeq U\Sigma^{1/N}, W_{N-1} \simeq \Sigma^{1/N}, \ldots, W_2 \simeq \Sigma^{1/N}, W_1 \simeq \Sigma^{1/N}V^\top$, where the symbol "$\simeq$" stands for equality up to zero-valued padding.[6] [7]*

The concept of balanced initialization, together with Theorem 1, leads to a guarantee for linear convergence (applicable to output dimension 1 — scalar regression) that holds with constant probability over the randomness in initialization:

**Theorem 2.** *For any constant $0 < p < 1/2$, there are constants $d_0', a > 0$ [8] such that the following holds. Assume $d_N = 1, d_0 \geq d_0'$, and that the weights $W_1(0), \ldots, W_N(0)$ are subject to balanced initialization (Procedure 1) such that the entries in $W_{1:N}(0)$ are independent zero-centered Gaussian perturbations with standard deviation $s \leq \|\Phi\|_2/\sqrt{ad_0^2}$. Suppose also that we run gradient*

---

[6] These assignments can be accomplished since $\min\{d_1, \ldots, d_{N-1}\} \geq \min\{d_0, d_N\}$.

[7] By design $W_{1:N} = A$ and $W_{j+1}^\top W_{j+1} = W_j W_j^\top$, $\forall j \in \{1, \ldots, N-1\}$ — these properties are actually all we need in Theorem 2, and step *(iii)* in Procedure 1 can be replaced by any assignment that meets them.

[8] As shown in the proof of the theorem (Appendix D.3), $d_0', a > 0$ can take on any pair of values for which: *(i)* $d_0' \geq 20$; and *(ii)* $\big(1 - 2\exp(-d_0'/16)\big)\big(3 - 4F(2/\sqrt{a/2})\big) \geq 2p$, where $F(\cdot)$ stands for the cumulative distribution function of the standard normal distribution. For example, if $p = 0.25$, it suffices to take any $d_0' \geq 100, a \geq 100$. We note that condition *(i)* here ($d_0' \geq 20$) serves solely for simplification of expressions in the theorem.

*descent with learning rate $\eta \le (s^2 d_0)^{4-2/N} / \left( 10^5 N^3 \|\Phi\|_2^{10-6/N} \right)$. Then, with probability at least $p$ over the random initialization, we have that for every $\epsilon > 0$ and:*

$$T \ge \frac{4}{\eta} \left( \ln(4) \left( \frac{\|\Phi\|_2}{s^2 d_0} \right)^{2-2/N} + \|\Phi\|_2^{2/N-2} \ln(\|\Phi\|_2^2/(8\epsilon)) \right),$$

*the loss at iteration $T$ of gradient descent — $\ell(T)$ — is no greater than $\epsilon$.*

*Proof.* See Appendix D.3. □

## 4 EXPERIMENTS

Balanced initialization (Procedure 1) possesses theoretical advantages compared with the customary layer-wise independent scheme — it allowed us to derive a convergence guarantee that holds with constant probability over the randomness of initialization (Theorem 2). In this section we present empirical evidence suggesting that initializing with balancedness may be beneficial in practice as well. For conciseness, some of the details behind our implementation are deferred to Appendix E.

We began by experimenting in the setting covered by our analysis — linear neural networks trained via gradient descent minimization of $\ell_2$ loss over whitened data. The dataset chosen for the experiment was UCI Machine Learning Repository's "Gas Sensor Array Drift at Different Concentrations" (Vergara et al., 2012; Rodriguez-Lujan et al., 2014). Specifically, we used the dataset's "Ethanol" problem — a scalar regression task with 2565 examples, each comprising 128 features (one of the largest numeric regression tasks in the repository). Starting with the customary initialization of layer-wise independent random Gaussian perturbations centered at zero, we trained a three layer network ($N = 3$) with hidden widths ($d_1, d_2$) set to 32, and measured the time (number of iterations) it takes to converge (reach training loss within $\epsilon = 10^{-5}$ from optimum) under different choices of standard deviation for the initialization. To account for the possibility of different standard deviations requiring different learning rates (values for $\eta$), we applied, for each standard deviation independently, a grid search over learning rates, and recorded the one that led to fastest convergence. The result of this test is presented in Figure 1(a). As can be seen, there is a range of standard deviations that leads to fast convergence (a few hundred iterations or less), below and above which optimization decelerates by orders of magnitude. This accords with our discussion at the end of Subsection 3.3, by which overly small initialization ensures approximate balancedness (small $\delta$; see Definition 1) but diminishes deficiency margin (small $c$; see Definition 2) — "vanishing gradient problem" — whereas large initialization hinders both approximate balancedness and deficiency margin — "exploding gradient problem". In that regard, as a sanity test for the validity of our analysis, in a case where approximate balancedness is met at initialization (small standard deviation), we measured its persistence throughout optimization. As Figure 1(c) shows, our theoretical findings manifest themselves here — trajectories of gradient descent indeed preserve weight balancedness.

In addition to a three layer network, we also evaluated a deeper, eight layer model (with hidden widths identical to the former — $N = 8$, $d_1 = \cdots = d_7 = 32$). In particular, using the same experimental protocol as above, we measured convergence time under different choices of standard deviation for the initialization. Figure 1(a) displays the result of this test alongside that of the three layer model. As the figure shows, transitioning from three layers to eight aggravated the instability with respect to initialization — there is now a narrow band of standard deviations that lead to convergence in reasonable time, and outside of this band convergence is extremely slow, to the point where it does not take place within the duration we allowed ($10^6$ iterations). From the perspective of our analysis, a possible explanation for the aggravation is as follows: under layer-wise independent initialization, the magnitude of the end-to-end matrix $W_{1:N}$ depends on the standard deviation in a manner that is exponential in depth, thus for large depths the range of standard deviations that lead to moderately sized $W_{1:N}$ (as required for a deficiency margin) is limited, and within this range, there may not be many standard deviations small enough to ensure approximate balancedness. The procedure of balanced initialization (Procedure 1) circumvents these difficulties — it assigns $W_{1:N}$ directly (no exponential dependence on depth), and distributes its content between the individual weights $W_1, \ldots, W_N$ in a perfectly balanced fashion. Rerunning the experiment of Figure 1(a) with this initialization replacing the customary layer-wise scheme (using same experimental protocol), we obtained the results shown in Figure 1(b) — both the original three layer network, and the deeper eight layer model, converged quickly under virtually all standard deviations tried.

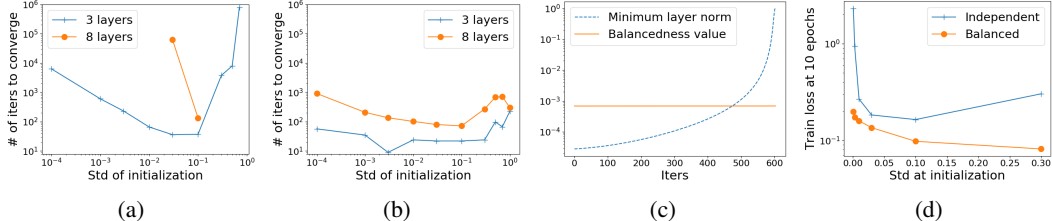

Figure 1: Experimental results. **(a)** Convergence of gradient descent training deep linear neural networks (depths 3 and 8) under customary initialization of layer-wise independent Gaussian perturbations with mean $0$ and standard deviation $s$. For each network, number of iterations required to reach $\epsilon = 10^{-5}$ from optimal training loss is plotted as a function of $s$ (missing values indicate no convergence within $10^6$ iterations). Dataset in this experiment is a numeric regression task from UCI Machine Learning Repository (details in text). Notice that fast convergence is attained only in a narrow band of values for $s$, and that this phenomenon is more extreme with the deeper network. **(b)** Same setup as in (a), but with layer-wise independent initialization replaced by balanced initialization (Procedure 1) based on Gaussian perturbations with mean $0$ and standard deviation $s$. Notice that this change leads to fast convergence, for both networks, under wide range of values for $s$. Notice also that the shallower network converges slightly faster, in line with the results of Saxe et al. (2014) and Arora et al. (2018) for $\ell_2$ loss. **(c)** For the run in (a) of a depth-3 network and standard deviation $s = 10^{-3}$, this plot shows degree of balancedness (minimal $\delta$ satisfying $\|W_{j+1}^\top W_{j+1} - W_j W_j^\top\|_F \leq \delta$, $\forall j \in \{1, \ldots, N-1\}$) against magnitude of weights ($\min_{j=1,\ldots,N} \|W_j W_j^\top\|_F$) throughout optimization. Notice that approximate balancedness persists under gradient descent, in line with our theoretical analysis. **(d)** Convergence of stochastic gradient descent training the fully-connected non-linear (ReLU) neural network of the MNIST tutorial built into TensorFlow (details in text). Customary layer-wise independent and balanced initializations — both based on Gaussian perturbations centered at zero — are evaluated, with varying standard deviations. For each configuration 10 epochs of optimization are run, followed by measurement of the training loss. Notice that although our theoretical analysis does not cover non-linear activation, softmax-cross-entropy loss and stochastic optimization, the conclusion of balanced initialization leading to improved convergence carries over to this setting.

As a final experiment, we evaluated the effect of balanced initialization in a setting that involves non-linear activation, softmax-cross-entropy loss and stochastic optimization (factors not accounted for by our analysis). For this purpose, we turned to the MNIST tutorial built into TensorFlow (Abadi et al., 2016),[9] which comprises a fully-connected neural network with two hidden layers (width $128$ followed by $32$) and ReLU activation (Nair and Hinton, 2010), trained through stochastic gradient descent (over softmax-cross-entropy loss) with batch size $100$, initialized via customary layer-wise independent Gaussian perturbations centered at zero. While keeping the learning rate at its default value $0.01$, we varied the standard deviation of initialization, and for each value measured the training loss after 10 epochs.[10] We then replaced the original (layer-wise independent) initialization with a balanced initialization based on Gaussian perturbations centered at zero (latter was implemented per Procedure 1, disregarding non-linear activation), and repeated the process. The results of this experiment are shown in Figure 1(d). Although our theoretical analysis does not cover non-linear activation, softmax-cross-entropy loss or stochasticity in optimization, its conclusion of balanced initialization leading to improved (faster and more stable) convergence carried over to such setting.

## 5 RELATED WORK

Theoretical study of gradient-based optimization in deep learning is a highly active area of research. As discussed in Section 1, a popular approach is to show that the objective landscape admits the properties of no poor local minima and strict saddle, which, by Ge et al. (2015); Lee et al. (2016); Panageas and Piliouras (2017), ensure convergence to global minimum. Many works, both classic (*e.g.* Baldi and Hornik (1989)) and recent (*e.g.* Choromanska et al. (2015); Kawaguchi (2016); Hardt and Ma (2016); Soudry and Carmon (2016); Haeffele and Vidal (2017); Nguyen and Hein (2017); Safran and Shamir (2018); Nguyen and Hein (2018); Laurent and Brecht (2018)), have focused on the validity of these properties in different deep learning settings. Nonetheless, to our knowledge,

---

[9]https://github.com/tensorflow/tensorflow/tree/master/tensorflow/examples/tutorials/mnist

[10]As opposed to the dataset used in our experiments with linear networks, measuring the training loss with MNIST is non-trivial computationally (involves passing through $60K$ examples). Therefore, rather than continuously polling training loss until it reaches a certain threshold, in this experiment we chose to evaluate speed of convergence by measuring the training loss once after a predetermined number of iterations.

the success of landscape-driven analyses in formally proving convergence to global minimum for a gradient-based algorithm, has thus far been limited to shallow (two layer) models only (*e.g.* Ge et al. (2016); Du and Lee (2018); Du et al. (2018a)).

An alternative to the landscape approach is a direct analysis of the trajectories taken by the optimizer. Various papers (*e.g.* Brutzkus and Globerson (2017); Li and Yuan (2017); Zhong et al. (2017); Tian (2017); Brutzkus et al. (2018); Li et al. (2018); Du et al. (2018c;b); Liao et al. (2018)) have recently adopted this strategy, but their analyses only apply to shallow models. In the context of linear neural networks, deep (three or more layer) models have also been treated — *cf.* Saxe et al. (2014) and Arora et al. (2018), from which we draw certain technical ideas for proving Lemma 1. However these treatments all apply to gradient flow (gradient descent with infinitesimally small learning rate), and thus do not formally address the question of computational efficiency.

To our knowledge, Bartlett et al. (2018) is the only existing work rigorously proving convergence to global minimum for a conventional gradient-based algorithm training a deep model. This work is similar to ours in the sense that it also treats linear neural networks trained via minimization of $\ell_2$ loss over whitened data, and proves linear convergence (to global minimum) for gradient descent. It is more limited in that it only covers the subclass of linear residual networks, *i.e.* the specific setting of uniform width across all layers ($d_0 = \cdots = d_N$) along with identity initialization. We on the other hand allow the input, output and hidden dimensions to take on any configuration that avoids "bottlenecks" (*i.e.* admits $\min\{d_1, \ldots, d_{N-1}\} \geq \min\{d_0, d_N\}$), and from initialization require only approximate balancedness (Definition 1), supporting many options beyond identity. In terms of the target matrix $\Phi$, Bartlett et al. (2018) treats two separate scenarios:[11] *(i)* $\Phi$ is symmetric and positive definite; and *(ii)* $\Phi$ is within distance $1/10e$ from identity.[12] Our analysis does not fully account for scenario *(i)*, which seems to be somewhat of a singularity, where all layers are equal to each other throughout optimization (see proof of Theorem 2 in Bartlett et al. (2018)). We do however provide a strict generalization of scenario *(ii)* — our assumption of deficiency margin (Definition 2), in the setting of linear residual networks, is met if the distance between target and identity is less than $0.5$.

## 6 CONCLUSION

For deep linear neural networks, we have rigorously proven convergence of gradient descent to global minima, at a linear rate, provided that the initial weight matrices are approximately balanced and the initial end-to-end matrix has positive deficiency margin. The result applies to networks with arbitrary depth, and any configuration of input/output/hidden dimensions that supports full rank, *i.e.* in which no hidden layer has dimension smaller than both the input and output.

Our assumptions on initialization — approximate balancedness and deficiency margin — are both necessary, in the sense that violating any one of them may lead to convergence failure, as we demonstrated explicitly. Moreover, for networks with output dimension 1 (scalar regression), we have shown that a balanced initialization, *i.e.* a random choice of the end-to-end matrix followed by a balanced partition across all layers, leads assumptions to be met, and thus convergence to take place, with constant probability. Rigorously proving efficient convergence with significant probability under customary layer-wise independent initialization remains an open problem. The recent work of Shamir (2018) suggests that this may not be possible, as at least in some settings, the number of iterations required for convergence is exponential in depth with overwhelming probability. This negative result, a theoretical manifestation of the "vanishing gradient problem", is circumvented by balanced initialization. Through simple experiments we have shown that the latter can lead to favorable convergence in deep learning practice, as it does in theory. Further investigation of balanced initialization, including development of variants for convolutional layers, is regarded as a promising direction for future research.

The analysis in this paper uncovers special properties of the optimization landscape in the vicinity of gradient descent trajectories. We expect similar ideas to prove useful in further study of gradient descent on non-convex objectives, including training losses of deep non-linear neural networks.

---

[11]There is actually an additional third scenario being treated — $\Phi$ is asymmetric and positive definite — but since that requires a dedicated optimization algorithm, it is outside our scope.

[12]$1/10e$ is the optimal (largest) distance that may be obtained (via careful choice of constants) from the proof of Theorem 1 in Bartlett et al. (2018).

ACKNOWLEDGMENTS

This work is supported by NSF, ONR, Simons Foundation, Schmidt Foundation, Mozilla Research, Amazon Research, DARPA and SRC. Nadav Cohen is a member of the Zuckerman Israeli Postdoctoral Scholars Program, and is supported by Schmidt Foundation.

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

## A $\ell_2$ LOSS OVER WHITENED DATA

Recall the $\ell_2$ loss of a linear predictor $W \in \mathbb{R}^{d_y \times d_x}$ as defined in Section 2:

$$L(W) = \frac{1}{2m} \|WX - Y\|_F^2 \,,$$

where $X \in \mathbb{R}^{d_x \times m}$ and $Y \in \mathbb{R}^{d_y \times m}$. Define $\Lambda_{xx} := \frac{1}{m} XX^\top \in \mathbb{R}^{d_x \times d_x}$, $\Lambda_{yy} := \frac{1}{m} YY^\top \in \mathbb{R}^{d_y \times d_y}$ and $\Lambda_{yx} := \frac{1}{m} YX^\top \in \mathbb{R}^{d_y \times d_x}$. Using the relation $\|A\|_F^2 = \text{Tr}(AA^\top)$, we have:

$$
\begin{aligned}
L(W) &= \tfrac{1}{2m} \text{Tr}\left((WX - Y)(WX - Y)^\top\right) \\
&= \tfrac{1}{2m} \text{Tr}(WXX^\top W^\top) - \tfrac{1}{m} \text{Tr}(WXY^\top) + \tfrac{1}{2m} \text{Tr}(YY^\top) \\
&= \tfrac{1}{2} \text{Tr}(W\Lambda_{xx}W^\top) - \text{Tr}(W\Lambda_{yx}^\top) + \tfrac{1}{2} \text{Tr}(\Lambda_{yy}) \,.
\end{aligned}
$$

By definition, when data is whitened, $\Lambda_{xx}$ is equal to identity, yielding:

$$
\begin{aligned}
L(W) &= \tfrac{1}{2} \text{Tr}(WW^\top) - \text{Tr}(W\Lambda_{yx}^\top) + \tfrac{1}{2} \text{Tr}(\Lambda_{yy}) \\
&= \tfrac{1}{2} \text{Tr}\left((W - \Lambda_{yx})(W - \Lambda_{yx})^\top\right) - \tfrac{1}{2} \text{Tr}(\Lambda_{yx}\Lambda_{yx}^\top) + \tfrac{1}{2} \text{Tr}(\Lambda_{yy}) \\
&= \tfrac{1}{2} \|W - \Lambda_{yx}\|_F^2 + c \,,
\end{aligned}
$$

where $c := -\frac{1}{2} \text{Tr}(\Lambda_{yx}\Lambda_{yx}^\top) + \frac{1}{2} \text{Tr}(\Lambda_{yy})$ does not depend on $W$. Hence we arrive at Equation (1).

## B APPROXIMATE BALANCEDNESS AND DEFICIENCY MARGIN UNDER CUSTOMARY INITIALIZATION

Two assumptions concerning initialization facilitate our main convergence result (Theorem 1): *(i)* the initial weights $W_1(0), \ldots, W_N(0)$ are approximately balanced (see Definition 1); and *(ii)* the initial end-to-end matrix $W_{1:N}(0)$ has positive deficiency margin with respect to the target $\Phi$ (see Definition 2). The current appendix studies the likelihood of these assumptions being met under customary initialization of random (layer-wise independent) Gaussian perturbations centered at zero.

For approximate balancedness we have the following claim, which shows that it becomes more and more likely the smaller the standard deviation of initialization is:

**Claim 2.** *Assume all entries in the matrices $W_j \in \mathbb{R}^{d_j \times d_{j-1}}$, $j = 1, \ldots, N$, are drawn independently at random from a Gaussian distribution with mean zero and standard deviation $s > 0$. Then, for any $\delta > 0$, the probability of $W_1, \ldots, W_N$ being $\delta$-balanced is at least $\max\{0, 1 - 10\delta^{-2}Ns^4d_{max}^3\}$, where $d_{max} := \max\{d_0, \ldots, d_N\}$.*

*Proof.* See Appendix D.4. □

In terms of deficiency margin, the claim below treats the case of a single output model (scalar regression), and shows that if the standard deviation of initialization is sufficiently small, with probability close to $0.5$, a deficiency margin will be met. However, for this deficiency margin to meet a chosen threshold $c$, the standard deviation need be sufficiently large.

**Claim 3.** *There is a constant $C_1 > 0$ such that the following holds. Consider the case where $d_N = 1$, $d_0 \geq 20$,[13] and suppose all entries in the matrices $W_j \in \mathbb{R}^{d_j \times d_{j-1}}$, $j = 1, \ldots, N$, are drawn independently at random from a Gaussian distribution with mean zero, whose standard deviation $s > 0$ is small with respect to the target, i.e. $s \leq \|\Phi\|_F^{1/N} / (10^5 d_0^3 d_1 \cdots d_{N-1} C_1)^{1/(2N)}$. Then, for any $c$ with $0 < c \leq \|\Phi\|_F / (10^5 d_0^3 C_1 (C_1 N)^{2N})$, the probability of the end-to-end matrix $W_{1:N}$ having deficiency margin $c$ with respect to $\Phi$ is at least $0.49$ if:[14][15]*

$$s \geq c^{1/(2N)} \cdot \left(C_1 N \|\Phi\|_F^{1/(2N)} / (d_1 \cdots d_{N-1})^{1/(2N)}\right) \,.$$

*Proof.* See Appendix D.5. □

---

[13]The requirement $d_0 \geq 20$ is purely technical, designed to simplify expressions in the claim.

[14]The probability $0.49$ can be increased to any $p < 1/2$ by increasing the constant $10^5$ in the upper bounds for $s$ and $c$.

[15]It is not difficult to see that the latter threshold is never greater than the upper bound for $s$, thus sought-after standard deviations always exist.

## C  CONVERGENCE FAILURES

In this appendix we show that the assumptions on initialization facilitating our main convergence result (Theorem 1) — approximate balancedness and deficiency margin — are both necessary, by demonstrating cases where violating each of them leads to convergence failure. This accords with widely observed empirical phenomena, by which successful optimization in deep learning crucially depends on careful initialization (*cf.* Sutskever et al. (2013)).

Claim 4 below shows[16] that if one omits from Theorem 1 the assumption of approximate balancedness at initialization, no choice of learning rate can guarantee convergence:

**Claim 4.** *Assume gradient descent with some learning rate $\eta > 0$ is a applied to a network whose depth $N$ is even, and whose input, output and hidden dimensions $d_0, \ldots, d_N$ are all equal to some $d \in \mathbb{N}$. Then, there exist target matrices $\Phi$ such that the following holds. For any $c$ with $0 < c < \sigma_{min}(\Phi)$, there are initializations for which the end-to-end matrix $W_{1:N}(0)$ has deficiency margin $c$ with respect to $\Phi$, and yet convergence will fail — objective will never go beneath a positive constant.*

*Proof.*  See Appendix D.6. □

In terms of deficiency margin, we provide (by adapting Theorem 4 in Bartlett et al. (2018)) a different, somewhat stronger result — there exist settings where initialization violates the assumption of deficiency margin, and despite being perfectly balanced, leads to convergence failure, for any choice of learning rate:[17]

**Claim 5.** *Consider a network whose depth $N$ is even, and whose input, output and hidden dimensions $d_0, \ldots, d_N$ are all equal to some $d \in \mathbb{N}$. Then, there exist target matrices $\Phi$ for which there are non-stationary initializations $W_1(0), \ldots, W_N(0)$ that are $0$-balanced, and yet lead gradient descent, under any learning rate, to fail — objective will never go beneath a positive constant.*

*Proof.*  See Appendix D.7. □

## D  DEFERRED PROOFS

We introduce some additional notation here in addition to the notation specified in Section 2. We use $\|A\|_\sigma$ to denote the spectral norm (largest singular value) of a matrix $A$, and sometimes $\|\mathbf{v}\|_2$ as an alternative to $\|\mathbf{v}\|$ — the Euclidean norm of a vector $\mathbf{v}$. Recall that for a matrix $A$, $vec(A)$ is its vectorization in column-first order. We let $F(\cdot)$ denote the cumulative distribution function of the standard normal distribution, *i.e.* $F(x) = \int_{-\infty}^{x} \frac{1}{\sqrt{2\pi}} e^{-\frac{1}{2}u^2} du$ ($x \in \mathbb{R}$).

To simplify the presentation we will oftentimes use $W$ as an alternative (shortened) notation for $W_{1:N}$ — the end-to-end matrix of a linear neural network. We will also use $L(\cdot)$ as shorthand for $L^1(\cdot)$ — the loss associated with a (directly parameterized) linear model, *i.e.* $L(W) := \frac{1}{2} \|W - \Phi\|_F^2$. Therefore, in the context of gradient descent training a linear neural network, the following expressions all represent the loss at iteration $t$:

$$\ell(t) = L^N(W_1(t), \ldots, W_N(t)) = L^1(W_{1:N}(t)) = L^1(W(t)) = L(W(t)) = \frac{1}{2} \|W(t) - \Phi\|_F^2 \, .$$

Also, for weights $W_j \in \mathbb{R}^{d_j \times d_{j-1}}$, $j = 1, \ldots, N$ of a linear neural network, we generalize the notation $W_{1:N}$, and define $W_{j:j'} := W_{j'} W_{j'-1} \cdots W_j$ for every $1 \le j \le j' \le N$. Note that $W_{j:j'}^\top = W_j^\top W_{j+1}^\top \cdots W_{j'}^\top$. Then, by a simple gradient calculation, the gradient descent updates (4) can be written as

$$W_j(t+1) = W_j(t) - \eta W_{j+1:N}^\top(t) \cdot \frac{dL}{dW}(W(t)) \cdot W_{1:j-1}^\top(t) \quad , 1 \le j \le N \, , \qquad (14)$$

where we define $W_{1:0}(t) := I_{d_0}$ and $W_{N+1:N}(t) := I_{d_N}$ for completeness.

---

[16]For simplicity of presentation, the claim treats the case of even depth and uniform dimension across all layers. It can easily be extended to account for arbitrary depth and input/output/hidden dimensions.

[17]This statement becomes trivial if one allows initialization at a suboptimal stationary point, *e.g.* $W_j(0) = 0$, $j = 1, \ldots, N$. Claim 5 rules out such trivialities by considering only non-stationary initializations.

Finally, recall the standard definition of the tensor product of two matrices (also known as the Kronecker product): for matrices $A \in \mathbb{R}^{m_A \times n_A}, B \in \mathbb{R}^{m_B \times n_B}$, their tensor product $A \otimes B \in \mathbb{R}^{m_A m_B \times n_A n_B}$ is defined as

$$A \otimes B = \begin{pmatrix} a_{1,1}B & \cdots & a_{1,n_A}B \\ \vdots & \ddots & \vdots \\ a_{m_A,1}B & \cdots & a_{m_A,n_A}B \end{pmatrix},$$

where $a_{i,j}$ is the element in the $i$-th row and $j$-th column of $A$.

## D.1 PROOF OF CLAIM 1

*Proof.* Recall that for any matrices $A$ and $B$ of compatible sizes $\sigma_{min}(A + B) \geq \sigma_{min}(A) - \sigma_{max}(B)$, and that the Frobenius norm of a matrix is always lower bounded by its largest singular value (Horn and Johnson (1990)). Using these facts, we have:

$$\sigma_{min}(W') = \sigma_{min}\big(\Phi + (W' - \Phi)\big) \geq \sigma_{min}(\Phi) - \sigma_{max}(W' - \Phi)$$
$$\geq \sigma_{min}(\Phi) - \|W' - \Phi\|_F \geq \sigma_{min}(\Phi) - \|W - \Phi\|_F$$
$$\geq \sigma_{min}(\Phi) - (\sigma_{min}(\Phi) - c) = c.$$

$\square$

## D.2 PROOF OF LEMMA 1

To prove Lemma 1, we will in fact prove a stronger result, Lemma 2 below, which states that for each iteration $t$, in addition to (9) being satisfied, certain other properties are also satisfied, namely: *(i)* the weight matrices $W_1(t), \ldots, W_N(t)$ are $2\delta$-balanced, and *(ii)* $W_1(t), \ldots, W_N(t)$ have bounded spectral norms.

**Lemma 2.** *Suppose the conditions of Theorem 1 are satisfied. Then for all $t \in \mathbb{N} \cup \{0\}$,*

$(\mathcal{A}(t))$ *For $1 \leq j \leq N - 1$, $\|W_{j+1}^\top(t)W_{j+1}(t) - W_j(t)W_j^\top(t)\|_F \leq 2\delta$.*

$(\mathcal{A}'(t))$ *If $t \geq 1$, then for $1 \leq j \leq N - 1$,*

$$\|W_{j+1}^\top(t)W_{j+1}(t) - W_j(t)W_j^\top(t)\|_F$$
$$\leq \|W_{j+1}^\top(t-1)W_{j+1}(t-1) - W_j(t-1)W_j^\top(t-1)\|_F$$
$$+\eta^2 \left\|\frac{dL^1}{dW}W(t-1)\right\|_F \cdot \left\|\frac{dL^1}{dW}W(t-1)\right\|_\sigma \cdot 4 \cdot (2\|\Phi\|_F)^{2(N-1)/N}.$$

$(\mathcal{B}(t))$ *If $t = 0$, then $\ell(t) \leq \frac{1}{2}\|\Phi\|_F^2$. If $t \geq 1$, then*

$$\ell(t) \leq \ell(t-1) - \frac{\eta}{2}\sigma_{min}(W(t-1))^{\frac{2(N-1)}{N}} \left\|\frac{dL^1}{dW}(W(t-1))\right\|_F^2.$$

$(\mathcal{C}(t))$ *For $1 \leq j \leq N$, $\|W_j(t)\|_\sigma \leq (4\|\Phi\|_F)^{1/N}$.*

First we observe that Lemma 1 is an immediate consequence of Lemma 2.

*Proof of Lemma 1.* Notice that condition $\mathcal{B}(t)$ of Lemma 2 for each $t \geq 1$ immediately establishes the conclusion of Lemma 1 at time step $t - 1$. $\square$

### D.2.1 PRELIMINARY LEMMAS

We next prove some preliminary lemmas which will aid us in the proof of Lemma 2. The first is a matrix inequality that follows from Lidskii's theorem. For a matrix $A$, let $\mathrm{Sing}(A)$ denote the rectangular diagonal matrix of the same size, whose diagonal elements are the singular values of $A$ arranged in non-increasing order (starting from the $(1, 1)$ position).

**Lemma 3** (Bhatia (1997), Exercise IV.3.5). *For any two matrices $A, B$ of the same size, $\|\operatorname{Sing}(A) - \operatorname{Sing}(B)\|_\sigma \le \|A - B\|_\sigma$ and $\|\operatorname{Sing}(A) - \operatorname{Sing}(B)\|_F \le \|A - B\|_F$.*

Using Lemma 3, we get:

**Lemma 4.** *Suppose $D_1, D_2 \in \mathbb{R}^{d \times d}$ are non-negative diagonal matrices with non-increasing values along the diagonal and $O \in \mathbb{R}^{d \times d}$ is an orthogonal matrix. Suppose that $\|D_1 - OD_2 O^\top\|_F \le \epsilon$, for some $\epsilon > 0$. Then:*

1. *$\|D_1 - OD_1 O^\top\|_F \le 2\epsilon$.*

2. *$\|D_1 - D_2\|_F \le \epsilon$.*

*Proof.* Since $D_1$ and $OD_2 O^T$ are both symmetric positive semi-definite matrices, their singular values are equal to their eigenvalues. Moreover, the singular values of $D_1$ are simply its diagonal elements and the singular values of $OD_2 O^T$ are simply the diagonal elements of $D_2$. Thus by Lemma 3 we get that $\|D_1 - D_2\|_F \le \|D_1 - OD_2 O^T\|_F \le \epsilon$. Since the Frobenius norm is unitarily invariant, $\|D_1 - D_2\|_F = \|OD_1 O^T - OD_2 O^T\|_F$, and by the triangle inequality it follows that

$$\|D_1 - OD_1 O^T\|_F \le \|OD_1 O^T - OD_2 O^T\|_F + \|D_1 - OD_2 O^T\|_F \le 2\epsilon.$$

$\square$

Lemma 5 below states that if $W_1, \ldots, W_N$ are approximately balanced matrices, *i.e.* $W_{j+1}^\top W_{j+1} - W_j W_j^\top$ has small Frobenius norm for $1 \le j \le N - 1$, then we can bound the Frobenius distance between $W_{1:j}^\top W_{1:j}$ and $(W_1^\top W_1)^j$ (as well as between $W_{j:N} W_{j:N}^\top$ and $(W_N W_N^\top)^{N-j+1}$).

**Lemma 5.** *Suppose that $d_N \le d_{N-1}, d_0 \le d_1$, and that for some $\nu > 0, M > 0$, the matrices $W_j \in \mathbb{R}^{d_j \times d_{j-1}}$, $1 \le j \le N$ satisfy, for $1 \le j \le N - 1$,*

$$\|W_{j+1}^\top W_{j+1} - W_j W_j^\top\|_F \le \nu, \tag{15}$$

*and for $1 \le j \le N$, $\|W_j\|_\sigma \le M$. Then, for $1 \le j \le N$,*

$$\|W_{1:j}^\top W_{1:j} - (W_1^\top W_1)^j\|_F \le \frac{3}{2}\nu \cdot M^{2(j-1)} j^2, \tag{16}$$

*and*

$$\|W_{j:N} W_{j:N}^\top - (W_N W_N^\top)^{N-j+1}\|_F \le \frac{3}{2}\nu \cdot M^{2(N-j)}(N - j + 1)^2. \tag{17}$$

*Moreover, if $\sigma_{min}$ denotes the minimum singular value of $W_{1:N}$, $\sigma_{1,min}$ denotes the minimum singular value of $W_1$ and $\sigma_{N,min}$ denotes the minimum singular value of $W_N$, then*

$$\sigma_{min}^2 - \frac{3}{2}\nu M^{2(N-1)} N^2 \le \begin{cases} \sigma_{N,min}^{2N} & : \quad d_N \ge d_0. \\ \sigma_{1,min}^{2N} & : \quad d_N \le d_0. \end{cases} \tag{18}$$

*Proof.* For $1 \le j \le N$, let us write the singular value decomposition of $W_j$ as $W_j = U_j \Sigma_j V_j^\top$, where $U_j \in \mathbb{R}^{d_j \times d_j}$ and $V_j \in \mathbb{R}^{d_{j-1} \times d_{j-1}}$ are orthogonal matrices and $\Sigma_j \in \mathbb{R}^{d_j \times d_{j-1}}$ is diagonal. We may assume without loss of generality that the singular values of $W_j$ are non-increasing along the diagonal of $\Sigma_j$. Then we can write (15) as

$$\|V_{j+1}\Sigma_{j+1}^\top \Sigma_{j+1} V_{j+1}^\top - U_j \Sigma_j \Sigma_j^\top U_j^\top\|_F \le \nu.$$

Since the Frobenius norm is invariant to orthogonal transformations, we get that

$$\|\Sigma_{j+1}^\top \Sigma_{j+1} - V_{j+1}^\top U_j \Sigma_j \Sigma_j^\top U_j^\top V_{j+1}\|_F \le \nu.$$

By Lemma 4, we have that $\|\Sigma_{j+1}^\top \Sigma_{j+1} - \Sigma_j \Sigma_j^\top\|_F \le \nu$ and $\|\Sigma_j \Sigma_j^\top - V_{j+1}^\top U_j \Sigma_j \Sigma_j^\top U_j^\top V_{j+1}\|_F \le 2\nu$. We may rewrite the latter of these two inequalities as

$$\|[\Sigma_j \Sigma_j^\top, V_{j+1}^\top U_j]\|_F = \|[\Sigma_j \Sigma_j^\top, V_{j+1}^\top U_j] U_j^\top V_{j+1}\|_F = \|\Sigma_j \Sigma_j^\top - V_{j+1}^\top U_j \Sigma_j \Sigma_j^\top U_j^\top V_{j+1}\|_F \le 2\nu.$$

Note that

$$W_{j:N} W_{j:N}^\top = W_{j+1:N} U_j \Sigma_j \Sigma_j^\top U_j^\top W_{j+1:N}^\top.$$

For matrices $A, B$, we have that $\|AB\|_F \le \|A\|_\sigma \cdot \|B\|_F$. Therefore, for $j+1 \le i \le N$, we have that

$$
\begin{aligned}
& \|W_{i:N}U_{i-1}(\Sigma_{i-1}\Sigma_{i-1}^\top)^{i-j}U_{i-1}^\top W_{i:N}^\top - W_{i+1:N}U_i(\Sigma_i\Sigma_i^\top)^{i-j+1}U_i^\top W_{i+1:N}^\top\|_F \\
=\ & \|W_{i+1:N}U_i\left(\Sigma_i V_i^\top U_{i-1}(\Sigma_{i-1}\Sigma_{i-1}^\top)^{i-j}U_{i-1}^\top V_i \Sigma_i^\top - (\Sigma_i\Sigma_i^\top)^{i-j+1}\right)U_i^\top W_{i+1:N}^\top\|_F \\
\le\ & \|W_{i+1:N}U_i\Sigma_i\|_\sigma^2 \cdot \|(\Sigma_{i-1}\Sigma_{i-1}^\top)^{i-j} + [V_i^\top U_{i-1}, (\Sigma_{i-1}\Sigma_{i-1}^\top)^{i-j}]U_{i-1}^\top V_i - (\Sigma_i^\top \Sigma_i)^{i-j}\|_F \\
\le\ & \|W_{i:N}\|_\sigma^2 \left(\|[V_i^\top U_{i-1}, (\Sigma_{i-1}\Sigma_{i-1}^\top)^{i-j}]\|_F + \|(\Sigma_{i-1}\Sigma_{i-1}^\top)^{i-j} - (\Sigma_i^\top \Sigma_i)^{i-j}\|_F\right).
\end{aligned}
$$

Next, we have that

$$
\begin{aligned}
\|[V_i^\top U_{i-1}, (\Sigma_{i-1}\Sigma_{i-1}^\top)^{i-j}]\|_F &\le \sum_{k=0}^{i-j-1} \|(\Sigma_{i-1}\Sigma_{i-1}^\top)^k[V_i^\top U_{i-1}, \Sigma_{i-1}\Sigma_{i-1}^\top](\Sigma_{i-1}\Sigma_{i-1}^\top)^{i-j-1-k}\|_F \\
&\le \sum_{k=0}^{i-j-1} \|(\Sigma_{i-1}\Sigma_{i-1}^\top)^{i-j-1}\|_\sigma \cdot \|[V_i^\top U_{i-1}, \Sigma_{i-1}\Sigma_{i-1}^\top]\|_F \\
&\le (i-j)\|W_{i-1}\|_\sigma^{2(i-j-1)} \cdot 2\nu.
\end{aligned}
$$

We now argue that $\|(\Sigma_{i-1}\Sigma_{i-1}^\top)^k - (\Sigma_i^\top \Sigma_i)^k\|_F \le \nu \cdot kM^{2(k-1)}$. Note that $\|\Sigma_{i-1}\Sigma_{i-1}^\top - \Sigma_i^\top \Sigma_i\|_F \le \nu$, verifying the case $k=1$. To see the general case, since square diagonal matrices commute, we have that

$$
\begin{aligned}
\|(\Sigma_{i-1}\Sigma_{i-1}^\top)^k - (\Sigma_i^\top \Sigma_i)^k\|_F &= \left\|(\Sigma_{i-1}\Sigma_{i-1}^\top - \Sigma_i^\top \Sigma_i) \cdot \left(\sum_{\ell=0}^{k-1}(\Sigma_{i-1}\Sigma_{i-1}^\top)^\ell(\Sigma_i^\top \Sigma_i)^{k-1-\ell}\right)\right\|_F \\
&\le \nu \cdot \sum_{\ell=0}^{k-1} \|W_{i-1}\|_\sigma^{2\ell} \cdot \|W_i\|_\sigma^{2(k-\ell-1)} \\
&\le \nu k M^{2(k-1)}.
\end{aligned}
$$

It then follows that

$$
\begin{aligned}
& \|W_{i:N}U_{i-1}(\Sigma_{i-1}\Sigma_{i-1}^\top)^{i-j}U_{i-1}^\top W_{i:N}^\top - W_{i+1:N}U_i(\Sigma_i\Sigma_i^\top)^{i-j+1}U_i^\top W_{i+1:N}^\top\|_F \\
\le\ & \|W_{i:N}\|_\sigma^2 \cdot \left((i-j)M^{2(i-j-1)} \cdot 2\nu + \nu(i-j)M^{2(i-j-1)}\right) \\
=\ & \|W_{i:N}\|_\sigma^2 \cdot 3\nu(i-j)M^{2(i-j-1)}.
\end{aligned}
$$

By the triangle inequality, we then have that

$$
\begin{aligned}
& \|W_{j:N}W_{j:N}^\top - U_N(\Sigma_N\Sigma_N^\top)^{N-j+1}U_N^\top\|_F \\
\le\ & \nu \sum_{i=j+1}^N \|W_{i:N}\|_\sigma^2 \cdot 3(i-j)M^{2(i-j-1)} \\
\le\ & 3\nu \sum_{i=j+1}^N (i-j)M^{2(N-i+1)}M^{2(i-j-1)} \\
=\ & 3\nu M^{2(N-j)} \sum_{i=j+1}^N (i-j) \le \frac{3}{2}\nu \cdot M^{2(N-j)} \cdot (N-j+1)^2.
\end{aligned}
\tag{19}
$$

By an identical argument (formally, by replacing $W_j$ with $W_{N-j+1}^\top$), we get that

$$
\|W_{1:j}^\top W_{1:j} - V_1(\Sigma_1^\top \Sigma_1)^j V_1^\top\|_F \le \frac{3}{2}\nu \cdot M^{2(j-1)} \cdot j^2.
\tag{20}
$$

(19) and (20) verify (17) and (16), respectively, so it only remains to verify (18).

Letting $j=1$ in (19), we get

$$
\|W_{1:N}W_{1:N}^\top - U_N(\Sigma_N\Sigma_N^\top)^N U_N^\top\|_F \le \frac{3}{2}\nu \cdot M^{2(N-1)} \cdot N^2.
\tag{21}
$$

Let us write the eigendecomposition of $W_{1:N}W_{1:N}^\top$ with an orthogonal eigenbasis as $W_{1:N}W_{1:N}^\top = U\Sigma U^\top$, where $\Sigma$ is diagonal with its (non-negative) elements arranged in non-increasing order and $U$ is orthogonal. We can write the left hand side of (21) as $\|U\Sigma U^\top - U_N(\Sigma_N\Sigma_N^\top)^N U_N^\top\|_F = \|\Sigma - U^\top U_N(\Sigma_N\Sigma_N^\top)^N U_N^\top U\|_F$.

By Lemma 4, we have that

$$\|\Sigma - (\Sigma_N\Sigma_N^\top)^N\|_F \le \frac{3}{2}\nu M^{2(N-1)}N^2. \tag{22}$$

Recall that $W \in \mathbb{R}^{d_N \times d_0}$. Suppose first that $d_N \le d_0$. Let $\sigma_{min}$ denote the minimum singular value of $W_{1:N}$ (so that $\sigma_{min}^2$ is the element in the $(d_N, d_N)$ position of $\Sigma \in \mathbb{R}^{d_N \times d_N}$), and $\sigma_{N,min}$ denote the minimum singular value (i.e. diagonal element) of $\Sigma_N$, which lies in the $(d_N, d_N)$ position of $\Sigma_N$. (Note that the $(d_N, d_N)$ position of $\Sigma_N \in \mathbb{R}^{d_N \times d_{N-1}}$ exists since $d_{N-1} \ge d_N$ by assumption.) Then

$$(\sigma_{N,min}^{2N} - \sigma_{min}^2)^2 \le \left(\frac{3}{2}\nu M^{2(N-1)}N^2\right)^2,$$

so

$$\sigma_{N,min}^{2N} \ge \sigma_{min}^2 - \frac{3}{2}\nu M^{2(N-1)}N^2.$$

By an identical argument using (20), we get that, in the case that $d_0 \le d_N$, if $\sigma_{1,min}$ denotes the minimum singular value of $\Sigma_1$, then

$$\sigma_{1,min}^{2N} \ge \sigma_{min}^2 - \frac{3}{2}\nu M^{2(N-1)}N^2.$$

(Notice that we have used the fact that the nonzero eigenvalues of $W_{1:N}W_{1:N}^\top$ are the same as the nonzero eigenvalues of $W_{1:N}^\top W_{1:N}$.) This completes the proof of (18). $\qquad\square$

Using Lemma 5, we next show in Lemma 6 that if $W_1, \ldots, W_N$ are approximately balanced, then an upper bound on $\|W_N \cdots W_1\|_\sigma$ implies an upper bound on $\|W_j\|_\sigma$ for $1 \le j \le N$.

**Lemma 6.** *Suppose $\nu, C$ are real numbers satisfying $C > 0$ and $0 < \nu \le \frac{C^{2/N}}{30N^2}$. Moreover suppose that the matrices $W_1, \ldots, W_N$ satisfy the following:*

   *1. For $1 \le j \le N-1$, $\|W_{j+1}^\top W_{j+1} - W_j W_j^\top\|_F \le \nu$.*

   *2. $\|W_N \cdots W_1\|_\sigma \le C$.*

*Then for $1 \le j \le N$, $\|W_j\|_\sigma \le C^{1/N} \cdot 2^{1/(2N)}$.*

*Proof.* For $1 \le j \le N$, let us write the singular value decomposition of $W_j$ as $W_j = U_j\Sigma_j V_j^\top$, where the singular values of $W_j$ are decreasing along the main diagonal of $\Sigma_j$. By Lemma 4, we have that for $1 \le j \le N-1$, $\|\Sigma_{j+1}^\top\Sigma_{j+1} - \Sigma_j\Sigma_j^\top\|_F \le \nu$, which implies that $\left|\|\Sigma_{j+1}^\top\Sigma_{j+1}\|_\sigma - \|\Sigma_j\Sigma_j^\top\|_\sigma\right| \le \nu$.

Write $M = \max_{1 \le j \le N}\|W_j\|_\sigma = \max_{1 \le j \le N}\|\Sigma_j\|_\sigma$. By the above we have that $\|\Sigma_j\Sigma_j^\top\|_\sigma \ge M^2 - N\nu$ for $1 \le j \le N$.

Let the singular value decomposition of $W_{1:N}$ be denoted by $W_{1:N} = U\Sigma V^\top$, so that $\|\Sigma\|_\sigma \le C$. Then by (17) of Lemma 5 and Lemma 4 (see also (22), where the same argument was used), we have that

$$\|\Sigma\Sigma^\top - (\Sigma_N\Sigma_N^\top)^N\|_F \le \frac{3}{2}\nu M^{2(N-1)}N^2.$$

Then

$$\|(\Sigma_N\Sigma_N^\top)^N\|_\sigma \le \|\Sigma\Sigma^\top\|_\sigma + \frac{3}{2}\nu M^{(2(N-1))}N^2 \le \|\Sigma\Sigma^\top\|_\sigma + \frac{3}{2}\nu\left(\|\Sigma_N\Sigma_N^\top\|_\sigma + \nu N\right)^{N-1}N^2. \tag{23}$$

Now recall that $\nu$ is chosen so that $\nu \le \frac{C^{2/N}}{30 \cdot N^2}$. Suppose for the purpose of contradiction that there is some $j$ such that $\|W_j W_j^\top\|_\sigma > 2^{1/N} C^{2/N}$. Then it must be the case that

$$\|\Sigma_N \Sigma_N^\top\|_\sigma > 2^{1/N} C^{2/N} - \nu \cdot N \ge (5/4)^{1/N} C^{2/N} > \nu \cdot 30 N^2, \tag{24}$$

where we have used that

$$2^{1/N} - (5/4)^{1/N} \ge \frac{1}{30N}$$

for all $N \ge 2$, which follows by considering the Laurent series $\exp(1/z) = \sum_{i=1}^\infty \frac{1}{i! z^i}$, which converges in $|z| > 0$ for $z \in \mathbb{C}$.

We now rewrite inequality (24) as

$$\nu \le \frac{\|\Sigma_N \Sigma_N^\top\|_\sigma}{30 N^2}. \tag{25}$$

Next, using (25) and $(1 + 1/x)^x \le e$ for all $x > 0$,

$$\frac{3}{2} \nu \left( \|\Sigma_N \Sigma_N^\top\|_\sigma + \nu N \right)^{N-1} N^2 \le \frac{e^{1/30}}{20} \cdot \|\Sigma_N \Sigma_N^\top\|_\sigma^N < \frac{e}{20} \cdot \|\Sigma_N \Sigma_N^\top\|_\sigma^N. \tag{26}$$

Since $\|(\Sigma_N \Sigma_N^\top)^N\|_\sigma = \|\Sigma_N \Sigma_N^\top\|_\sigma^N$, we get by combining (23) and (26) that

$$\|\Sigma_N \Sigma_N^\top\|_\sigma < (1 - e/20)^{-1/N} \cdot \|\Sigma \Sigma^\top\|_\sigma^{1/N} \le (1 - e/20)^{-1/N} \cdot C^{2/N},$$

and since $1 - e/20 > 1/(5/4)$, it follows that $\|\Sigma_N \Sigma_N^\top\|_\sigma < (5/4)^{1/N} C^{2/N}$, which contradicts (24). It follows that for all $1 \le j \le N$, $\|W_j W_j^\top\|_\sigma \le 2^{1/N} C^{2/N}$. The conclusion of the lemma then follows from the fact that $\|W_j W_j^\top\|_\sigma = \|W_j\|_\sigma^2$. □

### D.2.2 SINGLE-STEP DESCENT

Lemma 7 below states that if certain conditions on $W_1(t), \ldots, W_N(t)$ are met, the sought-after descent — Equation (9) — will take place at iteration $t$. We will later show (by induction) that the required conditions indeed hold for every $t$, thus the descent persists throughout optimization. The proof of Lemma 7 is essentially a discrete, single-step analogue of the continuous proof for Lemma 1 (covering the case of gradient flow) given in Section 3.

**Lemma 7.** *Assume the conditions of Theorem 1. Moreover, suppose that for some $t$, the matrices $W_1(t), \ldots, W_N(t)$ and the end-to-end matrix $W(t) := W_{1:N}(t)$ satisfy the following properties:*

1. *$\|W_j(t)\|_\sigma \le (4\|\Phi\|_F)^{1/N}$ for $1 \le j \le N$.*

2. *$\|W(t) - \Phi\|_\sigma \le \|\Phi\|_F$.*

3. *$\|W_{j+1}^\top(t) W_{j+1}(t) - W_j(t) W_j^\top(t)\|_F \le 2\delta$ for $1 \le j \le N-1$.*

4. *$\sigma_{min} := \sigma_{min}(W(t)) \ge c$.*

*Then, after applying a gradient descent update (4) we have that*

$$L(W(t+1)) - L(W(t)) \le -\frac{\eta}{2} \sigma_{min}^{2(N-1)/N} \left\| \frac{dL}{dW}(W(t)) \right\|_F^2.$$

*Proof.* For simplicity write $M = (4\|\Phi\|_F)^{1/N}$ and $B = \|\Phi\|_F$. We first claim that

$$\eta \le \min \left\{ \frac{1}{2M^{N-2}BN}, \frac{\sigma_{min}^{2(N-1)/N}}{24 \cdot 2M^{3N-4}N^2 B}, \frac{\sigma_{min}^{2(N-1)/N}}{24N^2 M^{4(N-1)}}, \frac{\sigma_{min}^{2(N-1)/(3N)}}{(24 \cdot 4M^{6N-8}N^4 B^2)^{1/3}} \right\}. \tag{27}$$

Since $c \le \sigma_{min}$, for (27) to hold it suffices to have

$$\eta \le \min \left\{ \frac{1}{8\|\Phi\|_F^{(2N-2)/N} N}, \frac{c^{2(N-1)/N}}{3 \cdot 2^{11} \|\Phi\|_F^{4(N-1)/N} N^2}, \frac{c^{2(N-1)/(3N)}}{3 \cdot 2^6 \left( \|\Phi\|_F^{(8N-8)/N} \right)^{1/3} N^{4/3}} \right\}.$$

As the minimum singular value of $\Phi$ must be at least $c$, we must have $c \leq \|\Phi\|_\sigma$. Since then $\frac{c}{\|\Phi\|_F} \leq \frac{c}{\|\Phi\|_\sigma} \leq 1$, it holds that

$$\frac{c^{2(N-1)/N}}{\|\Phi\|_F^{4(N-1)/N}} \leq \min\left\{ \frac{1}{\|\Phi\|_F^{2(N-1)/N}}, \frac{c^{2(N-1)/(3N)}}{\|\Phi\|_F^{(8N-8)/(3N)}} \right\},$$

meaning that it suffices to have

$$\eta \leq \frac{c^{2(N-1)/N}}{3 \cdot 2^{11} N^2 \|\Phi\|_F^{4(N-1)/N}},$$

which is guaranteed by (7).

Next, we claim that

$$
\begin{aligned}
2\delta &\leq \min\left\{ \frac{c^{2(N-1)/N}}{8 \cdot 2^4 N^3 \|\Phi\|_F^{2(N-2)/N}}, \frac{c^2}{6 \cdot 2^4 N^2 \|\Phi\|_F^{2(N-1)/N}} \right\} \\
&\leq \min\left\{ \frac{\sigma_{min}^{2(N-1)/N}}{8N^3 M^{2(N-2)}}, \frac{\sigma_{min}^2}{6N^2 M^{2(N-1)}} \right\}.
\end{aligned}
\tag{28}
$$

The second inequality above is trivial, and for the first to hold, since $c \leq \|\Phi\|_F$, it suffices to take

$$2\delta \leq \frac{c^2}{128 \cdot N^3 \cdot \|\Phi\|_F^{2(N-1)/N}},$$

which is guaranteed by the definition of $\delta$ in Theorem 1.

Next we continue with the rest of the proof. It follows from (14) that[18]

$$
\begin{aligned}
&W(t+1) - W(t) \\
&= \prod_1^{j=N} \left( W_j(t) - \eta W_{j+1:N}^\top(t)\frac{dL}{dW}(W(t))W_{1:j-1}^\top(t) \right) - W_{1:N}(t) \\
&= -\eta\left( \sum_{j=1}^N W_{j+1:N}W_{j+1:N}^\top(t)\frac{dL}{dW}(W(t))W_{1:j-1}^\top(t)W_{1:j-1}(t) \right) + (\star),
\end{aligned}
\tag{29}
$$

where $(\star)$ denotes higher order terms in $\eta$. We now bound the Frobenius norm of $(\star)$. To do this, note that since $L(W) = \frac{1}{2}\|W - \Phi\|_F^2$, $\frac{dL}{dW}(W(t)) = W(t) - \Phi$. Then

$$
\begin{aligned}
\|(\star)\|_F &\leq \sum_{k=2}^N \eta^k \cdot M^{k(N-1)+N-k} \cdot \left\|\frac{dL}{dW}(W(t))\right\|_F \cdot \left\|\frac{dL}{dW}(W(t))\right\|_\sigma^{k-1} \cdot \binom{N}{k} \\
&\leq \eta M^{2N-2}N \left\|\frac{dL}{dW}(W(t))\right\|_F \sum_{k=2}^N \left(\eta M^{N-2}BN\right)^{k-1} \\
&\leq \eta \cdot (2\eta M^{3N-4}N^2 B) \cdot \left\|\frac{dL}{dW}(W(t))\right\|_F,
\end{aligned}
\tag{30}
$$

---

[18]Here, for matrices $A_1, \ldots, A_K$ such that $A_K A_{K-1} \cdots A_1$ is defined, we write $\prod_1^{j=K} A_j := A_K A_{K-1} \cdots A_1$.

where the last inequality uses $\eta M^{N-2} BN \leq 1/2$, which is a consequence of (27). Next, by Lemma 5 with $\nu = 2\delta$,

$$
\left\| \sum_{j=1}^{N} W_{j+1:N} W_{j+1:N}^{\top}(t) \frac{dL}{dW}(W(t)) W_{1:j-1}^{\top}(t) W_{1:j-1}(t) \right.
$$

$$
\left. - \sum_{j=1}^{N} (W_N W_N^{\top})^{N-j} \frac{dL}{dW}(W(t))(W_1^{\top} W_1)^{j-1} \right\|_F
$$

$$
\leq \left\| \sum_{j=1}^{N} (W_{j+1:N} W_{j+1:N}^{\top}(t) - (W_N W_N^{\top})^{N-j}) \frac{dL}{dW}(W(t)) W_{1:j-1}^{\top}(t) W_{1:j-1}(t) \right\|_F
$$

$$
+ \left\| \sum_{j=1}^{N} (W_N W_N^{\top})^{N-j} \frac{dL}{dW}(W(t))(W_{1:j-1}^{\top} W_{1:j-1} - (W_1^{\top} W_1)^{j-1}) \right\|_F
$$

$$
\leq \left\| \frac{dL}{dW}(W(t)) \right\|_F \cdot \left( \sum_{j=1}^{N-1} \frac{3}{2} 2\delta \cdot M^{2(N-j)} (N-j)^2 M^{2(j-1)} + \sum_{j=2}^{N} \frac{3}{2} 2\delta \cdot M^{2(j-2)}(j-1)^2 M^{2(N-j)} \right)
$$

$$
\leq \left\| \frac{dL}{dW}(W(t)) \right\|_F \cdot 2\delta N^3 M^{2(N-2)}.
$$

Next, by standard properties of tensor product, we have that

$$
vec\left( \sum_{j=1}^{N} (W_N W_N^{\top})^{N-j} \frac{dL}{dW}(W(t))(W_1^{\top} W_1)^{j-1} \right)
$$

$$
= \sum_{j=1}^{N} \left( (W_1^{\top} W_1)^{j-1} \otimes (W_N W_N^{\top})^{N-j} \right) vec\left( \frac{dL}{dW}(W(t)) \right).
$$

Let us write eigenvalue decompositions $W_1^{\top} W_1 = UDU^{\top}, W_N W_N^{\top} = VEV^{\top}$. Then

$$
\sum_{j=1}^{N} \left( (W_1^{\top} W_1)^{j-1} \otimes (W_N W_N^{\top})^{N-j} \right)
$$

$$
= \sum_{j=1}^{N} \left( UD^{j-1} U^{\top} \otimes VE^{N-j} V^{\top} \right)
$$

$$
= (U \otimes V)\left( \sum_{j=1}^{N} D^{j-1} \otimes E^{N-j} \right)(U \otimes V)^{\top}
$$

$$
= O\Lambda O^{\top},
$$

with $O = U \otimes V$, and $\Lambda = \sum_{j=1}^{N} D^{j-1} \otimes E^{N-j}$. As $W_1 \in \mathbb{R}^{d_1 \times d_0}$, and $W_N \in \mathbb{R}^{d_N \times d_{N-1}}$, then $D \in \mathbb{R}^{d_0 \times d_0}, E \in \mathbb{R}^{d_N \times d_N}$, so $\Lambda \in \mathbb{R}^{d_0 d_N \times d_0 d_N}$. Moreover note that $\Lambda \succeq D^0 \otimes E^{N-1} + D^{N-1} \otimes E^0 = I_{d_0} \otimes E^{N-1} + D^{N-1} \otimes I_{d_N}$. If $\lambda_D$ denotes the minimum diagonal element of $D$ and $\lambda_E$ denotes the minimum diagonal element of $E$, then the minimum diagonal element of $\Lambda$ is therefore at least $\lambda_D^{N-1} + \lambda_E^{N-1}$. But, it follows from Lemma 5 (with $\nu = 2\delta$) that

$$
\max\{\lambda_D^N, \lambda_E^N\} \geq \sigma_{min}^2 - \frac{3}{2} 2\delta M^{2(N-1)} N^2 \geq 3\sigma_{min}^2/4,
$$

where the second inequality follows from (28). Hence the minimum diagonal element of $\Lambda$ is at least $(\sigma_{min}^2/(4/3))^{(N-1)/N} \geq \sigma_{min}^{2(N-1)/N}/(4/3)$.

It follows as a result of the above inequalities that if we write $E(t) = vec(W(t+1)) - vec(W(t)) + \eta(O\Lambda O^{\top})vec\left( \frac{dL}{dW}(W(t)) \right)$, then

$$
\|E(t)\|_2 = \left\| vec(W(t+1)) - vec(W(t)) + \eta(O\Lambda O^{\top})vec\left( \frac{dL}{dW}(W(t)) \right) \right\|_2
$$

$$
\leq \eta \left\| \frac{dL}{dW}(W(t)) \right\|_F \cdot (2\eta M^{3N-4} N^2 B + 2\delta N^3 M^{2(N-2)}).
$$

Then we have

$$
\begin{aligned}
&L(W(t+1)) - L(W(t)) \\
\leq\ & vec\left(\frac{d}{dW}L(W(t))\right)^\top vec\left(W(t+1) - W(t)\right) + \frac{1}{2}\|W(t+1) - W(t)\|_F^2 \\
=\ & \eta\left(-vec\left(\frac{d}{dW}L(W(t))\right)^\top (O\Lambda O^\top)vec\left(\frac{d}{dW}L(W(t))\right) + \frac{1}{\eta}vec\left(\frac{d}{dW}L(W(t))\right)^\top E(t)\right) \\
& + \frac{1}{2}\|W(t+1) - W(t)\|_F^2 \\
\leq\ & \eta\left(-\left\|\frac{d}{dW}L(W(t))\right\|_F^2 \cdot \frac{\sigma_{min}^{2(N-1)/N}}{4/3} + \left\|\frac{d}{dW}L(W(t))\right\|_F^2 \cdot \left(2\eta M^{3N-4}N^2 B + 2\delta N^3 M^{2(N-2)}\right)\right) \\
& + \frac{1}{2}\|W(t+1) - W(t)\|_F^2,
\end{aligned}
$$

where the first inequality follows since $L(W) = \frac{1}{2}\|W - \Phi\|_F^2$ is 1-smooth as a function of $W$.
Next, by (29) and (30),

$$
\begin{aligned}
&\|W(t+1) - W(t)\|_F^2 \\
\leq\ & 2\eta^2 \cdot \left(NM^{2(N-1)} \cdot \left\|\frac{dL}{dW}(W(t))\right\|_F\right)^2 + 2\eta^2 \cdot (2\eta M^{3N-4}N^2 B)^2 \cdot \left\|\frac{dL}{dW}(W(t))\right\|_F^2 \\
=\ & 2\eta^2 \left\|\frac{dL}{dW}(W(t))\right\|_F^2 \cdot \left(N^2 M^{4(N-1)} + (4\eta^2 M^{6N-8}N^4 B^2)\right).
\end{aligned}
\tag{31}
$$

Thus

$$
\begin{aligned}
&L(W(t+1)) - L(W(t)) \\
\leq\ & \eta \cdot \left\|\frac{dL}{dW}(W(t))\right\|_F^2 \cdot \left(-\frac{\sigma_{min}^{2(N-1)/N}}{4/3} + 2\eta M^{3N-4}N^2 B + 2\delta N^3 M^{2(N-2)}\right. \\
& \left. + \eta \cdot (N^2 M^{4(N-1)} + 4\eta^2 M^{6N-8}N^4 B^2)\right).
\end{aligned}
$$

By (27, 28), which bound $\eta, 2\delta$, respectively, we have that

$$
\begin{aligned}
&L(W(t+1)) - L(W(t)) \\
\leq\ & \eta \cdot \left\|\frac{dL}{dW}(W(t))\right\|_F^2 \cdot \left(-\frac{\sigma_{min}^{2(N-1)/N}}{4/3} + \frac{\sigma_{min}^{2(N-1)/N}}{24} + \frac{\sigma_{min}^{2(N-1)/N}}{8} + \frac{\sigma_{min}^{2(N-1)/N}}{24} + \frac{\sigma_{min}^{2(N-1)/N}}{24}\right) \\
=\ & -\frac{1}{2}\sigma_{min}^{2(N-1)/N}\eta \left\|\frac{dL}{dW}(W(t))\right\|_F^2.
\end{aligned}
\tag{32}
$$

$\square$

### D.2.3 PROOF OF LEMMA 2

*Proof of Lemma 2.* We use induction on $t$, beginning with the base case $t = 0$. Since the weights $W_1(0), \ldots, W_N(0)$ are $\delta$-balanced, we get that $\mathcal{A}(0)$ holds automatically. To establish $\mathcal{B}(0)$, note that since $W_{1:N}(0)$ has deficiency margin $c > 0$ with respect to $\Phi$, we must have $\|W_{1:N}(0) - \Phi\|_F \leq \sigma_{min}(\Phi) \leq \|\Phi\|_F$, meaning that $L^1(W_{1:N}(0)) \leq \frac{1}{2}\|\Phi\|_F^2$.

Finally, by $\mathcal{B}(0)$, which gives $\|W(0) - \Phi\|_F \leq \|\Phi\|_F$, we have that

$$
\|W(0)\|_\sigma \leq \|W(0)\|_F \leq \|W(0) - \Phi\|_F + \|\Phi\|_F \leq 2\|\Phi\|_F.
\tag{33}
$$

To show that the above implies $\mathcal{C}(0)$, we use condition $\mathcal{A}(0)$ and Lemma 6 with $C = 2\|\Phi\|_F$ and $\nu = 2\delta$. By the definition of $\delta$ in Theorem 1 and since $c \leq \|\Phi\|_F$, we have that

$$
2\delta \leq \frac{c^2}{128 \cdot N^3 \cdot \|\Phi\|_F^{2(N-1)/N}} = \frac{\|\Phi\|_F^{2/N}}{128 N^3} \cdot \frac{c^2}{\|\Phi\|_F^2} < \frac{\|\Phi\|_F^{2/N}}{30 N^2},
\tag{34}
$$

as required by Lemma 6. As $\mathcal{A}(0)$ and (33) verify the preconditions 1. and 2., respectively, of Lemma 6, it follows that for $1 \leq j \leq N$, $\|W_j(t)\|_\sigma \leq (2\|\Phi\|_F)^{1/N} \cdot 2^{1/(2N)} < (4\|\Phi\|_F)^{1/N}$, verifying $\mathcal{C}(0)$ and completing the proof of the base case.

The proof of Lemma 2 follows directly from the following inductive claims.

1. $\mathcal{A}(t), \mathcal{B}(t), \mathcal{C}(t) \Rightarrow \mathcal{B}(t+1)$. To prove this, we use Lemma 7. We verify first that the preconditions hold. First, $\mathcal{C}(t)$ immediately gives condition 1. of Lemma 7. By $\mathcal{B}(t)$, we have that $\|W(t) - \Phi\|_\sigma \leq \|W(t) - \Phi\|_F \leq \|\Phi\|_F$, giving condition 2. of Lemma 7. $\mathcal{A}(t)$ immediately gives condition 3. of Lemma 7. Finally, by $\mathcal{B}(t)$, we have that $L^N(W_1(t), \ldots, W_N(t)) \leq L^N(W_1(0), \ldots, W_N(0))$, so $\sigma_{min}(W_{1:N}(t)) \geq c$ by Claim 1. This verifies condition 4. of Lemma 7. Then Lemma 7 gives that $L^N(W_1(t+1), \ldots, W_N(t+1)) \leq L^N(W_1(t), \ldots, W_N(t)) - \frac{1}{2}\sigma_{min}(W(t))^{2(N-1)/N}\eta\|\frac{dL}{dW}(W(t))\|_F^2$, establishing $\mathcal{B}(t+1)$.

2. $\mathcal{A}(0), \mathcal{A}'(1), \ldots, \mathcal{A}'(t), \mathcal{A}(t), \mathcal{B}(0), \ldots, \mathcal{B}(t), \mathcal{C}(t) \Rightarrow \mathcal{A}(t+1), \mathcal{A}'(t+1)$. To prove this, note that for $1 \leq j \leq N-1$,

$$W_{j+1}^\top(t+1)W_{j+1}(t+1) - W_j(t+1)W_j^\top(t+1)$$
$$= \left(W_{j+1}^\top(t) - \eta W_{1:j}(t)\frac{dL}{dW}(W(t))^\top W_{j+2:N}(t)\right)$$
$$\cdot \left(W_{j+1}(t) - \eta W_{j+2:N}^\top(t)\frac{dL}{dW}(W(t))W_{1:j}^\top(t)\right)$$
$$- \left(W_j(t) - \eta W_{j+1:N}^\top(t)\frac{dL}{dW}(W(t))W_{1:j-1}^\top(t)\right)$$
$$\cdot \left(W_j^\top(t) - \eta W_{1:j-1}(t)\frac{dL}{dW}(W(t))^\top W_{j+1:N}(t)\right).$$

By $\mathcal{B}(0), \ldots, \mathcal{B}(t)$, $\|W_{1:N}(t) - \Phi\|_F \leq \|\Phi\|_F$. By the triangle inequality it then follows that $\|W_{1:N}(t)\|_\sigma \leq 2\|\Phi\|_F$. Also $\mathcal{A}(t)$ gives that for $1 \leq j \leq N-1$, $\|W_j(t)W_j^\top(t) - W_{j+1}^\top(t)W_{j+1}(t)\|_F \leq 2\delta$. By Lemma 6 with $C = 2\|\Phi\|_F, \nu = 2\delta$ (so that (34) is satisfied),

$$\left\|W_{j+1}^\top(t+1)W_{j+1}(t+1) - W_j(t+1)W_j^\top(t+1)\right\|_F$$
$$\leq \|W_{j+1}^\top(t)W_{j+1}(t) - W_j(t)W_j^\top(t)\|_F + \eta^2\left\|\frac{dL}{dW}(W(t))\right\|_F \cdot \left\|\frac{dL}{dW}(W(t))\right\|_\sigma$$
$$\cdot \left(\|W_{j+2:N}(t)\|_\sigma^2\|W_{1:j}(t)\|_\sigma^2 + \|W_{1:j-1}\|_\sigma^2\|W_{j+1:N}\|_\sigma^2\right)$$
$$\leq \|W_{j+1}^\top(t)W_{j+1}(t) - W_j(t)W_j^\top(t)\|_F$$
$$+ 4\eta^2\left\|\frac{dL}{dW}(W(t))\right\|_F\left\|\frac{dL}{dW}(W(t))\right\|_\sigma (2\|\Phi\|_F)^{2(N-1)/N}. \tag{35}$$

In the first inequality above, we have also used the fact that for matrices $A, B$ such that $AB$ is defined, $\|AB\|_F \leq \|A\|_\sigma\|B\|_F$. (35) gives us $\mathcal{A}'(t+1)$.

We next establish $\mathcal{A}(t+1)$. By $\mathcal{B}(i)$ for $0 \leq i \leq t$, we have that $\left\|\frac{dL}{dW}(W(i))\right\|_F = \|W - \Phi\|_F \leq \|\Phi\|_F$. Using $\mathcal{A}'(i)$ for $0 \leq i \leq t$ and summing over $i$ gives

$$\|W_{j+1}^\top(t+1)W_{j+1}(t+1) - W_j(t+1)W_j^\top(t+1)\|_F$$
$$\leq \|W_{j+1}^\top(0)W_{j+1}(0) - W_j(0)W_j^\top(0)\|_F$$
$$+ 4(2\|\Phi\|_F)^{2(N-1)/N} \cdot \eta^2\sum_{i=0}^t\left\|\frac{dL}{dW}(W(i))\right\|_F^2. \tag{36}$$

Next, by $\mathcal{B}(0), \ldots, \mathcal{B}(t)$, we have that $L(W(i)) \leq L(W(0))$ for $i \leq t$. Since $W(0)$ has deficiency margin of $c$ and by Claim 1, it then follows that $\sigma_{min}(W(i)) \geq c$ for all $i \leq t$.

Therefore, by summing $\mathcal{B}(0), \ldots, \mathcal{B}(t)$,

$$\frac{1}{2} c^{2(N-1)/N} \eta \sum_{i=0}^{t} \left\| \frac{dL}{dW} W(i) \right\|_F^2$$

$$\leq \quad \frac{1}{2} \eta \sum_{i=0}^{t} \sigma_{min}(W(i))^{2(N-1)/N} \left\| \frac{dL}{dW}(W(i)) \right\|_F^2$$

$$\leq \quad L(W(0)) - L(W(t))$$

$$\leq \quad L(W(0)) \leq \frac{1}{2} \|\Phi\|_F^2.$$

Therefore,

$$4 \left(2\|\Phi\|_F\right)^{2(N-1)/N} \eta^2 \sum_{i=0}^{t} \left\| \frac{dL}{dW} W(i) \right\|_F^2$$

$$\leq \quad 16 \|\Phi\|_F^{2(N-1)/N} \eta \frac{\|\Phi\|_F^2}{c^{2(N-1)/N}}$$

$$\leq \quad 16 \|\Phi\|_F^{2(N-1)/N} \cdot \frac{1}{3 \cdot 2^{11} \cdot N^3} \cdot \frac{c^{(4N-2)/N}}{\|\Phi\|_F^{(6N-4)/N}} \cdot \frac{\|\Phi\|_F^2}{c^{2(N-1)/N}} \qquad (37)$$

$$\leq \quad \frac{c^2}{256 N^3 \|\Phi\|_F^{2(N-1)/N}}$$

$$= \quad \delta,$$

where (37) follows from the definition of $\eta$ in (7), and the last equality follows from definition of $\delta$ in Theorem 1. By (36), it follows that

$$\|W_{j+1}^\top(t+1)W_{j+1}(t+1) - W_j(t+1)W_j^\top(t+1)\|_F \leq 2\delta,$$

verifying $\mathcal{A}(t+1)$.

3. $\mathcal{A}(t), \mathcal{B}(t) \Rightarrow \mathcal{C}(t)$. We apply Lemma 6 with $\nu = 2\delta$ and $C = 2\|\Phi\|_F$. First, the triangle inequality and $\mathcal{B}(t)$ give

$$\|W_{1:N}(t)\|_\sigma \leq \|\Phi\|_\sigma + \|\Phi - W_{1:N}(t)\|_\sigma \leq \|\Phi\|_F + \sqrt{2 \cdot L(W_{1:N}(t))} \leq 2\|\Phi\|_F,$$

verifying precondition 2. of Lemma 6. $\mathcal{A}(t)$ verifies condition 1. of Lemma 6, so for $1 \leq j \leq N$, $\|W_j(t)\|_\sigma \leq (4\|\Phi\|_F)^{1/N}$, giving $\mathcal{C}(t)$.

The proof of Lemma 2 then follows by induction on $t$. $\qquad \square$

### D.3 PROOF OF THEOREM 2

Theorem 2 is proven by combining Lemma 8 below, which implies that the balanced initialization is likely to lead to an end-to-end matrix $W_{1:N}(0)$ with sufficiently large deficiency margin, with Theorem 1, which establishes convergence.

**Lemma 8.** *Let $d \in \mathbb{N}, d \geq 20$; $b_2 > b_1 \geq 1$ be real numbers (possibly depending on $d$); and $\Phi \in \mathbb{R}^d$ be a vector. Suppose that $\mu$ is a rotation-invariant distribution[19] over $\mathbb{R}^d$ with a well-defined density, such that, for some $0 < \epsilon < 1$,*

$$\mathbb{P}_{V \sim \mu} \left[ \frac{\|\Phi\|_2}{\sqrt{b_2 d}} \leq \|V\|_2 \leq \frac{\|\Phi\|_2}{\sqrt{b_1 d}} \right] \geq 1 - \epsilon.$$

*Then, with probability at least $(1 - \epsilon) \cdot \frac{3 - 4F(2/\sqrt{b_1})}{2}$, $V$ will have deficiency margin $\|\Phi\|_2/(b_2 d)$ with respect to $\Phi$.*

---

[19]Recall that a distribution on vectors $V \in \mathbb{R}^d$ is *rotation-invariant* if the distribution of $V$ is the same as the distribution of $OV$, for any orthogonal $d \times d$ matrix $O$. If $V$ has a well-defined density, this is equivalent to the statement that for any $r > 0$, the distribution of $V$ conditioned on $\|V\|_2 = r$ is uniform over the sphere centered at the origin with radius $r$.

The proof of Lemma 8 is postponed to Appendix D.5, where Lemma 8 will be restated as Lemma 16.

One additional technique is used in the proof of Theorem 2, which leads to an improvement in the guaranteed convergence rate. Because the deficiency margin of $W_{1:N}(0)$ is very small, namely $\mathcal{O}(\|\Phi\|_2/d_0)$ (which is necessary for the theorem to maintain constant probability), at the beginning of optimization, $\ell(t)$ will decrease very slowly. However, after a certain amount of time, the deficiency margin of $W_{1:N}(t)$ will increase to a constant, at which point the decrease of $\ell(t)$ will be much faster. To capture this acceleration, we apply Theorem 1 a second time, using the larger deficiency margin at the new "initialization." From a geometric perspective, we note that the matrices $W_1(0), \ldots, W_N(0)$ are very close to 0, and the point at which $W_j(0) = 0$ for all $j$ is a saddle. Thus, the increase in $\ell(t) - \ell(t+1)$ over time captures the fact that the iterates $(W_1(t), \ldots, W_N(t))$ escape a saddle point.

*Proof of Theorem 2.* Choose some $a \geq 2$, to be specified later. By assumption, all entries of the end-to-end matrix at time 0, $W_{1:N}(0)$, are distributed as independent Gaussians of mean 0 and standard deviation $s \leq \|\Phi\|_2/\sqrt{ad_0^2}$. We will apply Lemma 8 to the vector $W_{1:N}(0) \in \mathbb{R}^{d_0}$. Since its distribution is obviously rotation-invariant, in remains to show that the distribution of the norm $\|W_{1:N}(0)\|_2$ is not too spread out. The following lemma — a direct consequence of the Chernoff bound applied to the $\chi^2$ distribution with $d_0$ degrees of freedom — will give us the desired result:

**Lemma 9** (Laurent and Massart (2000), Lemma 1). *Suppose that $d \in \mathbb{N}$ and $V \in \mathbb{R}^d$ is a vector whose entries are i.i.d. Gaussians with mean 0 and standard deviation $s$. Then, for any $k > 0$,*

$$\mathbb{P}\left[\|V\|_2^2 \geq s^2\left(d + 2k + 2\sqrt{kd}\right)\right] \leq \exp(-k)$$
$$\mathbb{P}\left[\|V\|_2^2 \leq s^2\left(d - 2\sqrt{kd}\right)\right] \leq \exp(-k).$$

By Lemma 9 with $k = d_0/16$, we have that

$$\mathbb{P}\left[\frac{s^2 d_0}{2} \leq \|V\|_2^2 \leq 2s^2 d_0\right] \geq 1 - 2\exp(-d_0/16).$$

We next use Lemma 8, with $b_1 = \|\Phi\|_2^2/(2s^2 d_0^2), b_2 = 2\|\Phi\|_2^2/(s^2 d_0^2)$; note that since $a \geq 2$, $b_1 \geq 1$, as required by the lemma. Lemma 8 then implies that with probability at least

$$(1 - 2\exp(-d_0/16))\frac{3 - 4F\left(2/\sqrt{a/2}\right)}{2}, \tag{38}$$

$W_{1:N}(0)$ will have deficiency margin $s^2 d_0/2\|\Phi\|_2$ with respect to $\Phi$. By the definition of balanced initialization (Procedure 1) $W_1(0), \ldots, W_N(0)$ are 0-balanced. Since $2^4 \cdot 6144 < 10^5$, our assumption on $\eta$ gives

$$\eta \leq \frac{(s^2 d_0)^{4-2/N}}{2^4 \cdot 6144 N^3 \|\Phi\|_2^{10-6/N}}, \tag{39}$$

so that Equation (7) holds with $c = \frac{s^2 d_0}{2\|\Phi\|_2}$. The conditions of Theorem 1 thus hold with probability at least that given in Equation (38). In such a constant probability event, by Theorem 1 (and the fact that a positive deficiency margin implies $L^1(W_{1:N}(0)) \leq \frac{1}{2}\|\Phi\|_2^2$), if we choose

$$t_0 \geq \eta^{-1}\left(\frac{2\|\Phi\|_2}{s^2 d_0}\right)^{2-2/N}\ln(4), \tag{40}$$

then $L^1(W_{1:N}(t_0)) \leq \frac{1}{8}\|\Phi\|_2^2$, meaning that $\|W_{1:N}(t_0) - \Phi\|_2 \leq \frac{1}{2}\|\Phi\|_2 = \|\Phi\|_2 - \frac{1}{2}\sigma_{min}(\Phi)$. Moreover, by condition $\mathcal{A}(t_0)$ of Lemma 2 and the definition of $\delta$ in Theorem 1, we have, for $1 \leq j \leq N - 1$,

$$\|W_{j+1}^T(t_0)W_{j+1}(t_0) - W_j(t_0)W_j^T(t_0)\|_F \leq \frac{2s^4 d_0^2}{(2\|\Phi\|_2)^2 \cdot 256 N^3 \|\Phi\|_2^{2-2/N}} = \frac{s^4 d_0^2}{512 N^3 \|\Phi\|_2^{4-2/N}}. \tag{41}$$

We now apply Theorem 1 again, verifying its conditions again, this time with the initialization $(W_1(t_0), \ldots, W_N(t_0))$. First note that the end-to-end matrix $W_{1:N}(t_0)$ has deficiency margin $c = \|\Phi\|_2/2$ as shown above. The learning rate $\eta$, by Equation (39), satisfies Equation (7) with $c = \|\Phi\|_2/2$. Finally, since

$$\frac{s^4 d_0^2}{512 N^3 \|\Phi\|_2^{4-2/N}} \leq \frac{\|\Phi\|^{2/N}}{(a^2 d_0^2) \cdot 512 N^3} \leq \frac{\|\Phi\|^{2/N}(1/2)^2}{256 N^3}$$

for $d_0 \geq 2$, by Equation (41), the matrices $W_1(t_0), \ldots, W_N(t_0)$ are $\delta$-balanced with $\delta = \frac{\|\Phi\|^{2/N}(1/2)^2}{256 N^3}$. Iteration $t_0$ thus satisfies the conditions of Theorem 1 with deficiency margin $\|\Phi\|_2/2$, meaning that for

$$T - t_0 \geq \eta^{-1} \cdot 2^{2-2/N} \cdot \|\Phi\|^{2/N-2} \ln\left(\frac{\|\Phi\|_2^2}{8\epsilon}\right), \tag{42}$$

we will have $\ell(T) \leq \epsilon$. Therefore, by Equations (40) and (42), to ensure that $\ell(T) \leq \epsilon$, we may take

$$T \geq 4\eta^{-1}\left(\ln(4)\left(\frac{\|\Phi\|_2}{s^2 d_0}\right)^{2-2/N} + \|\Phi\|_2^{2/N-2}\ln(\|\Phi\|_2^2/(8\epsilon))\right).$$

Recall that this entire analysis holds only with the probability given in Equation (38). As $\lim_{d\to\infty}(1 - 2\exp(-d/16)) = 1$ and $\lim_{a\to\infty}(3 - 4F(2\sqrt{2/a}))/2 = 1/2$, for any $0 < p < 1/2$, there exist $a, d_0' > 0$ such that for $d_0 \geq d_0'$, the probability given in Equation (38) is at least $p$. This completes the proof.

$\square$

In the context of the above proof, we remark that the expressions $1 - 2\exp(-d_0/16)$ and $(3 - 4F(2\sqrt{2/a}))/2$ converge to their limits of $1$ and $1/2$, respectively, as $d_0, a \to \infty$ quite quickly. For instance, to obtain a probability of greater than $0.25$ of the initialization conditions being met, we may take $d_0 \geq 100, a \geq 100$.

### D.4 Proof of Claim 2

We first consider the probability of $\delta$-balancedness holding between any two layers:

**Lemma 10.** *Suppose $a, b, d \in \mathbb{N}$ and $A \in \mathbb{R}^{a \times d}, B \in \mathbb{R}^{d \times b}$ are matrices whose entries are distributed as i.i.d. Gaussians with mean 0 and standard deviation $s$. Then for $k \geq 1$,*

$$\mathbb{P}\left[\left\|A^T A - BB^T\right\|_F \geq k s^2 \sqrt{2d(a+b)^2 + d^2(a+b)}\right] \leq 1/k^2. \tag{43}$$

*Proof.* Note that for $1 \leq i, j \leq d$, let $X_{ij}$ be the random variable $(A^T A - BB^T)_{ij}$, so that

$$X_{ij} = (A^T A - BB^T)_{ij} = \sum_{1 \leq \ell \leq a} A_{\ell i} A_{\ell j} - \sum_{1 \leq r \leq b} B_{ir} B_{jr}.$$

If $i \neq j$, then

$$\mathbb{E}[X^2] = \sum_{1 \leq \ell \leq a} \mathbb{E}[A_{\ell i}^2 A_{\ell j}^2] + \sum_{1 \leq r \leq b} \mathbb{E}[B_{ir}^2 B_{jr}^2] = (a+b)s^4.$$

We next note that for a normal random variable $Y$ of variance $s^2$ and mean 0, $\mathbb{E}[Y^4] = 3s^4$. Then if $i = j$,

$$\mathbb{E}[X^2] = s^4 \cdot (3(a+b) + a(a-1) + b(b-1) - ab) \leq s^4((a+b)^2 + 2(a+b)).$$

Thus

$$\begin{aligned}
\mathbb{E}[\|A^T A - BB^T\|_F^2] &\leq s^4(d((a+b)^2 + 2(a+b)) + d(d-1)(a+b)) \\
&\leq s^4(2d(a+b)^2 + d^2(a+b)).
\end{aligned}$$

Then (43) follows from Markov's inequality. $\square$

Now the proof of Claim 2 follows from a simple union bound:

*Proof of Claim 2.* By (43) of Lemma 10, for each $1 \leq j \leq N-1$, $k \geq 1$,

$$\mathbb{P}\left[\|W_{j+1}^T W_{j+1} - W_j W_j^T\|_F \geq ks^2\sqrt{10d_{max}^3}\right] \leq 1/k^2.$$

By the union bound,

$$\mathbb{P}\left[\forall 1 \leq j \leq N-1, \ \ \|W_{j+1}^T W_{j+1} - W_j W_j^T\|_F \leq ks^2\sqrt{10d_{max}^3}\right] \geq 1 - N/k^2,$$

and the claim follows with $\delta = ks^2\sqrt{10d_{max}^3}$. $\hfill\square$

## D.5 PROOF OF CLAIM 3

We begin by introducing some notation. Given $d \in \mathbb{N}$ and $r > 0$, we let $B^d(r)$ denote the open ball of radius $r$ centered at the origin in $\mathbb{R}^d$. For an open subset $U \subset \mathbb{R}^d$, let $\partial U := \bar{U}\backslash U$ be its boundary, where $\bar{U}$ denotes the closure of $U$. For the special case of $U = B^d(r)$, we will denote by $S^d(r)$ the boundary of such a ball, i.e. the sphere of radius $r$ centered at the origin in $\mathbb{R}^d$. Let $S^d := S^d(1)$ and $B^d := B^d(1)$. There is a well-defined uniform (Haar) measure on $S^d(r)$ for all $d, r$, which we denote by $\sigma^{d,r}$; we assume $\sigma^{d,r}$ is normalized so that $\sigma^{d,r}(S^d(r)) = 1$. Finally, since in the context of this claim we have $d_N = 1$, we allow ourselves to regard the end-to-end matrix $W_{1:N} \in \mathbb{R}^{1 \times d_0}$ as both a matrix and a vector.

To establish Claim 3, we will use the following low-degree anti-concentration result of Carbery and Wright (2001) (see also Lovett (2010); Meka et al. (2016)):

**Lemma 11** (Carbery and Wright (2001)). *There is an absolute constant $C_0$ such that the following holds. Suppose that $h$ is a multilinear polynomial of $K$ variables $X_1, \ldots, X_K$ and of degree $N$. Suppose that $X_1, \ldots, X_K$ are i.i.d. Gaussian. Then, for any $\epsilon > 0$:*

$$\mathbb{P}\left[|h(X_1, \ldots, X_K)| \leq \epsilon \cdot \sqrt{\mathrm{Var}[h(X_1, \ldots, X_K)]}\right] \leq C_0 N \epsilon^{1/N}.$$

The below lemma characterizes the norm of the end-to-end matrix $W_{1:N}$ following zero-centered Gaussian initialization:

**Lemma 12.** *For any constant $0 < C_2 < 1$, there is an absolute constant $C_1 > 0$ such that the following holds. Let $N, d_0, \ldots, d_{N-1} \in \mathbb{N}$. Set $d_N = 1$. Suppose that for $1 \leq j \leq N$, $W_j \in \mathbb{R}^{d_j \times d_{j-1}}$ are matrices whose entries are i.i.d. Gaussians of standard deviation $s$ and mean 0. Then*

$$\mathbb{P}\left[s^{2N}d_1\cdots d_{N-1}\left(\frac{1}{C_1 N}\right)^{2N} \leq \|W_{1:N}\|_2^2 \leq C_1 d_0^2 d_1 \cdots d_{N-1}s^{2N}\right] \geq C_2.$$

*Proof.* Let $f(W_1, \ldots, W_N) = \|W_{1:N}\|_2^2$, so that $f$ is a polynomial of degree $2N$ in the entries of $W_1, \ldots, W_N$. Notice that

$$f(W_1, \ldots, W_N) = \sum_{i_0=1}^{d_0}\left(\sum_{i_1=1}^{d_1}\cdots\sum_{i_{N-1}=1}^{d_{N-1}}(W_N)_{1,i_{N-1}}(W_{N-1})_{i_{N-1},i_{N-2}}\cdots(W_1)_{i_1,i_0}\right)^2.$$

For $1 \leq i_0 \leq d_0$, set

$$g_{i_0}(W_1, \ldots, W_N) = \sum_{i_1=1}^{d_1}\cdots\sum_{i_{N-1}=1}^{d_{N-1}}(W_N)_{1,i_{N-1}}(W_{N-1})_{i_{N-1},i_{N-2}}\cdots(W_1)_{i_1,i_0},$$

so that $f = \sum_{i_0=1}^{d_0} g_{i_0}^2$. Since each $g_{i_0}$ is a multilinear polynomial in $W_1, \ldots, W_N$, we have that $\mathbb{E}[g_{i_0}(W_1, \ldots, W_N)] = 0$ for all $1 \leq i_0 \leq d_0$. Also

$$\begin{aligned}
\mathrm{Var}[g_{i_0}(W_1, \ldots, W_N)] &= \mathbb{E}[g_{i_0}(W_1, \ldots, W_N)^2]\\
&= \sum_{i_1=1}^{d_1}\cdots\sum_{i_{N-1}=1}^{d_{N-1}}\mathbb{E}\left[(W_N)_{1,i_{N-1}}^2(W_{N-1})_{i_{N-1},i_{N-2}}^2\cdots(W_1)_{i_1,i_0}^2\right]\\
&= d_1 d_2 \cdots d_{N-1}s^{2N}.
\end{aligned}$$

It then follows by Markov's inequality that for any $k \geq 1$, $\mathbb{P}[g_{i_0}^2 \geq k s^{2N} d_1 \cdots d_{N-1}] \leq 1/k$. For any constant $B_1$ (whose exact value will be specified below), it follows that

$$
\begin{aligned}
& \mathbb{P}[f(W_1, \ldots, W_N) \geq B_1 d_0^2 d_1 d_2 \cdots d_{N-1} s^{2N}] \\
= \ & \mathbb{P}\left[\sum_{i_0=1}^{d_0} g_{i_0}(W_1, \ldots, W_N)^2 \geq B_1 d_0^2 d_1 d_2 \cdots d_{N-2} s^{2N}\right] \\
\leq \ & d_0 \cdot \mathbb{P}[g_1(W_1, \ldots, W_N)^2 \geq B_1 d_0 d_1 \cdots d_{N-1} s^{2N}] \\
\leq \ & 1/B_1.
\end{aligned}
\tag{44}
$$

Next, by Lemma 11, there is an absolute constant $C_0 > 0$ such that for any $\epsilon > 0$, and any $1 \leq i_0 \leq d_0$,

$$
\mathbb{P}\left[|g_{i_0}(W_1, \ldots, W_N)| \leq \epsilon^N \sqrt{s^{2N} d_1 \cdots d_{N-1}}\right] \leq C_0 N \epsilon.
$$

Since $f^2 \geq g_{i_0}^2$ for each $i_0$, it follows that

$$
\mathbb{P}[f(W_1, \ldots, W_N) \geq \epsilon^{2N} s^{2N} d_1 \cdots d_{N-1}] \geq 1 - C_0 N \epsilon.
\tag{45}
$$

Next, given $0 < C_2 < 1$, choose $\epsilon = (1 - C_2)/(2C_0 N)$, and $B_1 = 2/(1 - C_2)$. Then by (44) and (45) and a union bound, we have that

$$
\mathbb{P}\left[\left(\frac{1 - C_2}{2C_0 N}\right)^{2N} s^{2N} d_1 \cdots d_{N-1} \leq f(W_1, \ldots, W_N) \leq \frac{2}{1 - C_2} s^{2N} d_0^2 d_1 \cdots d_{N-1}\right] \geq C_2.
$$

The result of the lemma then follows by taking $C_1 = \max\left\{\frac{2}{1 - C_2}, \frac{2C_0}{1 - C_2}\right\}$. $\qquad\square$

**Lemma 13.** *Let $N, d_0, \ldots, d_{N-1} \in \mathbb{N}$, and set $d_N = 1$. Suppose $W_j \in \mathbb{R}^{d_j \times d_{j-1}}$ for $1 \leq j \leq N$, are matrices whose entries are i.i.d. Gaussians with mean $0$ and standard deviation $s$. Then, the distribution of $W_{1:N}$ is rotation-invariant.*

*Proof.* First we remark that for any orthogonal matrix $O \in \mathbb{R}^{d_0 \times d_0}$, the distribution of $W_1$ is the same as that of $W_1 O$. To see this, let us denote the rows of $W_1$ by $(W_1)_1, \ldots, (W_1)_{d_1}$, and the columns of $O$ by $O^1, \ldots, O^{d_0}$. Then the $(i_1, i_0)$ entry of $W_1 O$, for $1 \leq i_1 \leq d_1, 1 \leq i_0 \leq d_0$ is $\langle (W_1)_{i_1}, O^{i_0} \rangle$, which is a Gaussian with mean $0$ and standard deviation $s$, since $\|O^{i_0}\|_2 = 1$. Since $\langle O^{i_0}, O^{i_0'} \rangle = 0$ for $i_0 \neq i_0'$, the covariance between any two distinct entries of $W_1 O$ is $0$. Therefore, the entries of $W_1 O$ are independent Gaussians with mean $0$ and standard deviation $s$, just as are the entries of $W_1$.

But now for any matrix $O \in \mathbb{R}^{d_0 \times d_0}$, the distribution of $W_{1:N} O$ is the distribution of $W_N W_{N-1} \cdots W_2 (W_1 O)$, which is the same as the distribution of $W_N W_{N-1} \cdots W_2 W_1 = W_{1:N}$, since $W_1, W_2, \ldots, W_N$ are all independent. $\qquad\square$

For a dimension $d \in \mathbb{N}$, radius $r > 0$, and $0 < h < r$, a *$(d, r)$-hyperspherical cap of height $h$* is a subset $\mathcal{C} \subset B^d(r)$ of the form $\{x \in B^d(r) : \langle x, u \rangle \geq r - h\}$, where $u$ is any $d$-dimensional unit vector. We define the *area of a $(d, r)$-hyperspherical cap of height $h$ — $\mathcal{C}$ — to be $\sigma^{d,r}(\partial \mathcal{C} \cap S^d(r))$.

**Lemma 14.** *For $d \geq 20$, choose any $0 \leq h \leq 1$. Then, the area of a $(d, 1)$-hyperspherical cap of height $h$ is at least*

$$
\frac{3 - 4F((1 - h)\sqrt{d - 3})}{2}.
$$

*Proof.* In Chudnov (1986), it is shown that the area of a $(d, 1)$-hyperspherical cap of height $h$ is given by $\frac{1 - C_{d-2}(h)/C_{d-2}(0)}{2}$, where

$$
C_d(h) := \int_0^{1-h} (1 - t^2)^{(d-1)/2} dt.
$$

Next, by the inequality $1 - t^2 \geq \exp(-2t^2)$ for $0 \leq t \leq 1/2$,

$$
\begin{aligned}
\int_0^1 (1-t^2)^{(d-3)/2} dt &\geq \int_0^{1/2} \exp\left(2 \cdot \frac{-t^2(d-3)}{2}\right) dt \\
&= \sqrt{\pi/(d-3)} \cdot \frac{2F(\sqrt{(d-3)/2}) - 1}{2} \\
&\geq \sqrt{\pi/(d-3)} \cdot \frac{1 - 2\exp(-(d-3)/4)}{2},
\end{aligned}
\tag{46}
$$

where the last inequality follows from the standard estimate $F(x) \geq 1 - \exp(-x^2/2)$ for $x \geq 1$. Also, since $1 - t^2 \leq \exp(-t^2)$ for all $t$,

$$
\begin{aligned}
\int_0^{1-h} (1-t^2)^{(d-3)/2} dt &\leq \int_0^{1-h} \exp\left(\frac{-t^2(d-3)}{2}\right) dt \\
&= \sqrt{2\pi/(d-3)} \cdot \frac{2F((1-h)\sqrt{d-3}) - 1}{2}.
\end{aligned}
\tag{47}
$$

Therefore, for $d \geq 20$, by (46) and (47),

$$
\begin{aligned}
\frac{1 - C_{d-2}(h)/C_{d-2}(0)}{2} &\geq \frac{1 - \frac{\sqrt{2} \cdot (2F((1-h)\sqrt{d-3}) - 1)}{1 - 2\exp(-(d-3)/4)}}{2} \\
&\geq \frac{1 - \sqrt{2} \cdot (2F((1-h)\sqrt{d-3}) - 1) \cdot (1 + 4\exp(-(d-3)/4))}{2} \\
&\geq \frac{3 - 4F((1-h)\sqrt{d-3})}{2},
\end{aligned}
$$

where the second inequality has used $1/(1-y) \leq 1 + 2y$ for all $0 < y < 1/2$ (and where $y = 2\exp((-(d-3)/4)) < 2\exp(-17/4) < 1/2$), and the final inequality uses $1 + 4\exp(-(d-3)/4) \leq \sqrt{2}$ for $d \geq 20$. The above chain of inequalities gives us the desired result. $\square$

**Lemma 15.** *Let $d \in \mathbb{N}, d \geq 20$; $a \geq 1$ be a real number (possibly depending on $d$); and $\Phi \in \mathbb{R}^d$ be some vector. Set $r = \|\Phi\|_2/\sqrt{ad}$, and suppose that $V \in S^d(r)$ is drawn according to the uniform measure. Then, with probability at least $\frac{3 - 4F(2/\sqrt{a})}{2}$, $V$ will have deficiency margin $\|\Phi\|_2/(ad)$ with respect to $\Phi$.*

*Proof.* By rescaling, we may assume without loss of generality that $\|\Phi\|_2 = 1$, so that $r = 1/\sqrt{ad}$. Let $\mathcal{D}$ denote the intersection of $B^d(r)$ with the open $d$-ball of radius $1 - 1/(ad)$ centered at $\Phi$. Let $\mathcal{C} \subset B^d(r)$ denote the $(d, r)$-hyperspherical cap of height $r \cdot (1 - 2/(\sqrt{ad})) = r - 2/(ad)$ whose base is orthogonal to the line between $\mathbf{0}$ and $\Phi$ (see Figure 2). Note that $\sigma^{d,r}(\partial \mathcal{D} \cap S^d(r))$, the Haar measure of the portion of $\partial \mathcal{D}$ intersecting $S^d(r)$, gives the probability that $V$ belongs to the boundary of $\mathcal{D}$. By Lemma 14 above (along with rescaling arguments), since $d \geq 20$, $\sigma^{d,r}(\partial \mathcal{C} \cap S^d(r)) \geq \frac{1}{2} \cdot (3 - 4F(2/\sqrt{a}))$, and therefore $V \in \partial \mathcal{C}$ with at least this probability.

We next claim that $\mathcal{C} \subseteq \mathcal{D}$. To see this, first let $\mathcal{T} \subset \mathbb{R}^d$ denote the $(d-1)$-sphere of radius $1 - 1/(ad)$ centered at $\Phi$ (see Figure 2). Let $P$ be the intersection of $\mathcal{T}$ with the line from $\mathbf{0}$ to $\Phi$, and $Q$ denote the intersection of this line with the unique hyperplane of codimension 1 containing $\mathcal{T} \cap \partial B^d(r)$ — we denote this hyperplane by $\mathcal{H}$. If we can show that $\|P - Q\|_2 \leq 1/(ad)$, then it follows that $\mathcal{C}$ lies entirely on the other side of $\mathcal{H}$ as $\mathbf{0}$, which will complete the proof that $\mathcal{C} \subseteq \mathcal{D}$.

The calculation of $\|P - Q\|_2$ is simply an application of the law of cosines: letting $\theta$ be the angle determining the intersection of $\partial B^d(r)$ and $\mathcal{T}$ (see Figure 2), note that

$$
(1 - 1/(ad))^2 = r^2 + 1^2 - 2r\cos\theta = 1/(ad) + 1 - 2/\sqrt{ad} \cdot \cos(\theta),
$$

so

$$
d(P, Q) = r\cos\theta - 1/(ad) = \frac{1}{2}(1/(ad) - 1/(a^2 d^2)) < 1/(ad),
$$

as desired.

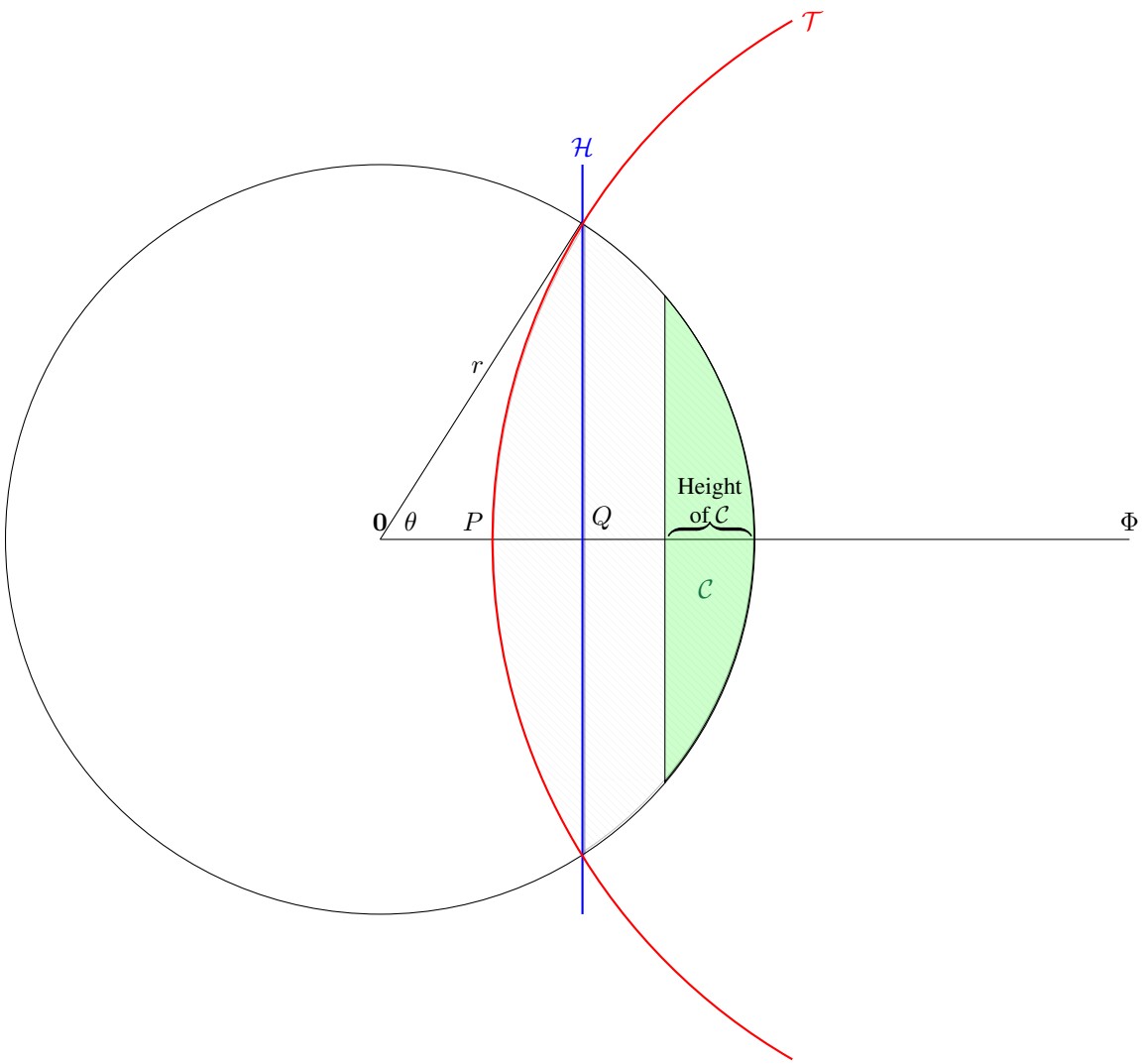

Figure 2: Figure for proof of Lemma 15. The dashed region denotes $\mathcal{D}$. Not to scale.

Using that $\mathcal{C} \subseteq \mathcal{D}$, we continue with the proof. Notice the fact that $\mathcal{C} \subseteq \mathcal{D}$ is equivalent to $\partial \mathcal{C} \cap S^d(r) \subseteq \partial \mathcal{D} \cap S^d(r)$, by the structure of $\mathcal{C}$ and $\mathcal{D}$. Since the probability that $V$ lands in $\partial \mathcal{C}$ is at least $\frac{3 - 4F(2/\sqrt{a})}{2}$, this lower bound applies to $V$ landing in $\partial \mathcal{D}$ as well. Since all $V \in \partial \mathcal{D}$ have distance at most $1 - 1/(ad)$ from $\Phi$, and since $\sigma_{min}(\Phi) = \|\Phi\|_2 = 1$, it follows that for any $V \in \partial \mathcal{D}$, $\|V - \Phi\|_2 \le \sigma_{min}(\Phi) - 1/(ad)$. Therefore, with probability of at least $\frac{3 - 4F(2/\sqrt{a})}{2}$, $V$ has deficiency margin $\|\Phi\|_2/(ad)$ with respect to $\Phi$. $\qquad\square$

**Lemma 16** (Lemma 8 restated). *Let $d \in \mathbb{N}, d \ge 20$; $b_2 > b_1 \ge 1$ be real numbers (possibly depending on d); and $\Phi \in \mathbb{R}^d$ be a vector. Suppose that $\mu$ is a rotation-invariant distribution over $\mathbb{R}^d$ with a well-defined density, such that, for some $0 < \epsilon < 1$,*

$$\mathbb{P}_{V \sim \mu} \left[ \frac{\|\Phi\|_2}{\sqrt{b_2 d}} \le \|V\|_2 \le \frac{\|\Phi\|_2}{\sqrt{b_1 d}} \right] \ge 1 - \epsilon.$$

*Then, with probability at least $(1 - \epsilon) \cdot \frac{3 - 4F(2/\sqrt{b_1})}{2}$, $V$ will have deficiency margin $\|\Phi\|_2/(b_2 d)$ with respect to $\Phi$.*

*Proof.* By rescaling we may assume that $\|\Phi\|_2 = 1$ without loss of generality. Then the deficiency margin of $V$ is equal to $1 - \|V - \Phi\|_2$. $\mu$ has a well-defined density, so we can set $\hat{\mu}$ to be the

probability density function of $\|V\|_2$. Since $\mu$ is rotation-invariant, we can integrate over spherical coordinates, giving

$$
\begin{aligned}
&\mathbb{P}[1 - \|V - \Phi\|_2 \geq 1/(b_2 d)] \\
&= \int_0^\infty \mathbb{P}\big[1 - \|V - \Phi\|_2 \geq 1/(b_2 d) \,\big|\, \|V\|_2 = r\big] \hat{\mu}(r) dr \\
&\geq \int_{1/(\sqrt{b_2 d})}^{1/(\sqrt{b_1 d})} \frac{3 - 4F(2r\sqrt{d})}{2} \hat{\mu}(r) dr \\
&\geq \frac{3 - 4F(2/\sqrt{b_1})}{2} \cdot \int_{1/(\sqrt{b_2 d})}^{1/(\sqrt{b_1 d})} \hat{\mu}(r) dr \\
&\geq \frac{3 - 4F(2/\sqrt{b_1})}{2} \cdot (1 - \epsilon),
\end{aligned}
$$

where the first inequality used Lemma 15 and the fact that the distribution of $V$ conditioned on $\|V\|_2 = r$ is uniform on $S^d(r)$. $\qquad\square$

Now we are ready to prove Claim 3:

*Proof of Claim 3.* We let $W \in \mathbb{R}^{1 \times d_0} \simeq \mathbb{R}^{d_0}$ denote the random vector $W_{1:N}$; also let $\mu$ denote the distribution of $W$, so that by Lemma 13, $\mu$ is rotation-invariant. Let $C_1$ be the constant from Lemma 12 for $C_2 = 999/1000$. For some $a \geq 10^5$, the standard deviation of the entries of each $W_j$ is given by

$$
s = \left( \frac{\|\Phi\|_2^2}{a d_0^3 d_1 \cdots d_{N-1} C_1} \right)^{1/(2N)}. \tag{48}
$$

Then by Lemma 12,

$$
\mathbb{P}\left[ \frac{\|\Phi\|_2^2}{a d_0^3 C_1} \cdot \left( \frac{1}{C_1 N} \right)^{2N} \leq \|W\|_2^2 \leq \frac{\|\Phi\|_2^2}{a d_0} \right] \geq \frac{999}{1000}.
$$

Then Lemma 16, with $d = d_0$, $b_1 = a$ and $b_2 = a d_0^2 C_1 \cdot (C_1 N)^{2N}$, implies that with probability at least $\frac{999}{1000} \cdot \frac{3 - 4F(2/\sqrt{a})}{2}$, $W$ has deficiency margin $\|\Phi\|_2/(a d_0^3 C_1^{2N+1} N^{2N})$ with respect to $\Phi$. But $a \geq 10^5$ implies that this probability is at least $0.49$, and from (48),

$$
\frac{\|\Phi\|_2}{a d_0^3 C_1^{2N+1} N^{2N}} = \frac{s^{2N} d_1 \cdots d_{N-1}}{\|\Phi\|_2 (C_1 N)^{2N}}. \tag{49}
$$

Next recall the assumption in the hypothesis that $s \geq C_1 N (c \cdot \|\Phi\|_2 / (d_1 \cdots d_{N-1}))^{1/2N}$. Then the deficiency margin in (49) is at least

$$
\frac{\big( C_1 N (c \|\Phi\|_2 / (d_1 \cdots d_{N-1}))^{1/(2N)} \big)^{2N} d_1 \cdots d_{N-1}}{\|\Phi\|_2 (C_1 N)^{2N}} = c,
$$

completing the proof.

$\qquad\square$

## D.6 PROOF OF CLAIM 4

*Proof.* The target matrices $\Phi$ that will be used to prove the claim satisfy $\sigma_{min}(\Phi) = 1$. We may assume without loss of generality that $c \geq 3/4$, the reason being that if a matrix has deficiency margin $c$ with respect to $\Phi$ and $c' < c$, it certainly has deficiency margin $c'$ with respect to $\Phi$.

We first consider the case $d = 1$, so that the target and all matrices are simply real numbers; we will make a slight abuse of notation in identifying $1 \times 1$ matrices with their unique entries. We set $\Phi = 1$. For all choices of $\eta$, we will set the initializations $W_1(0), \ldots, W_N(0)$ so that $W_{1:N}(0) = c$. Then

$$
\|W_{1:N}(0) - \Phi\|_F = |W_{1:N}(0) - \Phi| = 1 - c = \sigma_{min}(\Phi) - c,
$$

so the initial end-to-end matrix $W_{1:N}(0) \in \mathbb{R}^{1 \times 1}$ has deficiency margin $c$. Now fix $\eta$. Choose $A \in \mathbb{R}$ with

$$A = \max \left\{ \sqrt{\eta N}, \frac{2}{\eta(1-c)c^{(N-1)/N}}, 2000, 20/\eta, \left( \frac{20 \cdot 10^{2N-1}}{\eta^{2N}} \right)^{1/(2N-2)} \right\}. \quad (50)$$

We will set:

$$W_j(0) = \begin{cases} Ac^{1/N} & : & 1 \le j \le N/2 \\ c^{1/N}/A & : & N/2 < j \le N, \end{cases} \quad (51)$$

so that $W_{1:N}(0) = c$. Then since $L^N(W_1, \ldots, W_N) = \frac{1}{2}(1 - W_N \cdots W_1)^2$, the gradient descent updates are given by

$$W_j(t+1) = W_j(t) - \eta(W_{1:N}(t) - 1) \cdot W_{1:j-1}(t)W_{j+1:N}(t),$$

where we view $W_1(t), \ldots, W_N(t)$ as real numbers. This gives

$$W_j(1) = \begin{cases} c^{1/N}A - \eta(c-1)c^{(N-1)/N}/A & : & 1 \le j \le N/2 \\ c^{1/N}/A - \eta(c-1)c^{(N-1)/N}A & : & N/2 < j \le N. \end{cases}$$

Since $3/4 \le c < 1$ and $-\eta(c-1)c^{(N-1)/N}A \ge 0$, we have that $A/2 \le 3A/4 \le W_j(1)$ for $1 \le j \le N/2$. Next, since $\frac{1-c}{1-c^{1/N}} \le N$ for $0 \le c < 1$, we have that $A^2 \ge \eta N \ge \frac{\eta(1-c)}{1-c^{1/N}}$, which implies that $A^2 \ge c^{1/N}A^2 + \eta(1-c)$, or $c^{1/N}A + \frac{\eta(1-c)}{A} \le A$. Thus $W_j(1) \le A$ for $N/2 < j \le N$. Similarly, using the same bound $3/4 \le c < 1$ and the fact that $\eta(1-c)c^{(N-1)/N}A \ge 2$ we get $\frac{3}{16}\eta A \le W_j(1) \le \eta A$ for $N/2 < j \le N$. In particular, for all $1 \le j \le N$, we have that $\frac{\min\{\eta,1\}}{10}A \le W_j(1) \le \max\{\eta, 1\}A$.

We prove the following lemma by induction:

**Lemma 17.** *For each $t \ge 1$, the real numbers $W_1(t), \ldots, W_N(t)$ all have the same sign and this sign alternates for each integer $t$. Moreover, there are real numbers $2 \le B(t) < C(t)$ for $t \ge 1$ such that for $1 \le j \le N$, $B(t) \le |W_j(t)| \le C(t)$ and $\eta B(t)^{2N-1} \ge 20C(t)$.*

*Proof.* First we claim that we may take $B(1) = \frac{\min\{\eta,1\}}{10}A$ and $C(1) = \max\{\eta, 1\}A$. We have shown above that $B(1) \le W_j(1) \le C(1)$ for all $j$. Next we establish that $\eta B(1)^{2N-1} \ge 20C(1)$. If $\eta \le 1$, then

$$\eta B(1)^{2N-1} = \eta^{2N} \cdot (A/10)^{2N-1} \ge 20A = 20C(1),$$

where the inequality follows from $A \ge \left( \frac{20 \cdot 10^{2N-1}}{\eta^{2N}} \right)^{1/(2N-2)}$ by definition of $A$. If $\eta \ge 1$, then

$$\eta B(1)^{2N-1} = \eta(A/10)^{2N-1} \ge 20\eta A = 20C(1),$$

where the inequality follows from $A \ge 2000 \ge \left( 20 \cdot 10^{2N-1} \right)^{1/(2N-2)}$ by definition of $A$.

Now, suppose the statement of Lemma 17 holds for some $t$. Suppose first that $W_j(t)$ are all positive for $1 \le j \le N$. Then for all $j$, as $B(t) \ge 2$, and $\eta B(t)^{2N-1} \ge 20C(t)$,

$$\begin{aligned} W_j(t+1) &\le C(t) - \eta \cdot (B(t)^N - 1) \cdot B(t)^{N-1} \\ &\le C(t) - \frac{\eta}{2}B(t)^{2N-1} \\ &\le -9C(t), \end{aligned}$$

which establishes that $W_j(t+1)$ is negative for all $j$. Moreover,

$$\begin{aligned} W_j(t+1) &\ge -\eta(C(t)^N - 1) \cdot C(t)^{N-1} \\ &\ge -\eta C(t)^{2N-1}. \end{aligned}$$

Now set $B(t+1) = 9C(t)$ and $C(t+1) = \eta C(t)^{2N-1}$. Since $N \ge 2$, we have that

$$\eta B(t+1)^{2N-1} = \eta(9C(t))^{2N-1} \ge \eta 9^3 C(t)^{2N-1} > 20\eta C(t)^{2N-1} = 20C(t+1).$$

The case that all $W_j(t)$ are negative for $1 \le j \le N$ is nearly identical, with the same values for $B(t+1), C(t+1)$ in terms of $B(t), C(t)$, except all $W_j(t+1)$ will be positive. This establishes the inductive step and completes the proof of Lemma 17. □

By Lemma 17, we have that for all $t \geq 1$, $L^N(W_1(t), \ldots, W_N(t)) = \frac{1}{2}(W_{1:N}(t) - 1)^2 \geq \frac{1}{2}(2^N - 1)^2 > 0$, thus completing the proof of Claim 4 for the case where all dimensions are equal to 1.

For the general case where $d_0 = d_1 = \cdots = d_N = d$ for some $d \geq 1$, we set $\Phi = I_d$, and given $c, \eta$, we set $W_j(0)$ to be the $d \times d$ diagonal matrix where all diagonal entries except the first one are equal to 1, and where the first diagonal entry is given by Equation (51), where $A$ is given by Equation (50). It is easily verified that all entries of $W_j(t)$, $1 \leq j \leq N$, except for the first diagonal element of each matrix, will remain constant for all $t \geq 0$, and that the first diagonal elements evolve exactly as in the 1-dimensional case presented above. Therefore the loss in the $d$-dimensional case is equal to the loss in the 1-dimensional case, which is always greater than some positive constant. $\square$

We remark that the proof of Claim 4 establishes that the loss $\ell(t) := L^N(W_1(t), \ldots, W_N(t))$ grows at least exponentially in $t$ for the chosen initialization. Such behavior, in which gradients and weights explode, indeed takes place in deep learning practice if initialization is not chosen with care.

### D.7   PROOF OF CLAIM 5

*Proof.* We will show that a target matrix $\Phi \in \mathbb{R}^{d \times d}$ which is symmetric with at least one negative eigenvalue, along with identity initialization ($W_j(0) = I_d, \forall j \in \{1, \ldots, N\}$), satisfy the conditions of the claim. First, note that non-stationarity of initialization is met, as for any $1 \leq j \leq N$,

$$\frac{\partial L^N(W_1(0), \ldots, W_N(0))}{\partial W_j(0)} = W_{j+1:N}(0)^\top (W_{1:N}(0) - \Phi)W_{1:j-1}(0) = I_d - \Phi \neq \mathbf{0},$$

where the last inequality follows since $\Phi$ has a negative eigenvalue. To analyze gradient descent we use the following result, which was established in Bartlett et al. (2018):

**Lemma 18** (Bartlett et al. (2018), Lemma 6). *If $W_1(0), \ldots, W_N(0)$ are all initialized to identity, $\Phi$ is symmetric, $\Phi = UDU^\top$ is a diagonalization of $\Phi$, and gradient descent is performed with any learning rate, then for each $t \geq 0$ there is a diagonal matrix $\hat{D}(t)$ such that $W_j(t) = U\hat{D}(t)U^\top$ for each $1 \leq j \leq N$.*

By Lemma 18, for any choice of learning rate $\eta$, the end-to-end matrix at time $t$ is given by $W_{1:N}(t) = U\hat{D}(t)^N U^\top$. As long as some diagonal element of $D$ is negative, say equal to $-\lambda < 0$, then

$$\ell(t) = L^N(W_1(t), \ldots, W_N(t)) = \frac{1}{2}\|W_{1:N}(t) - \Phi\|_F^2 = \frac{1}{2}\|\hat{D}(t)^L - D\|_F^2 \geq \frac{1}{2}\lambda^2 > 0.$$

$\square$

## E   IMPLEMENTATION DETAILS

Below we provide implementation details omitted from our experimental report (Section 4).

The platform used for running the experiments is PyTorch (Paszke et al., 2017). For compliance with our analysis, we applied PCA whitening to the numeric regression dataset from UCI Machine Learning Repository. That is, all instances in the dataset were preprocessed by an affine operator that ensured zero mean and identity covariance matrix. Subsequently, we rescaled labels such that the uncentered cross-covariance matrix $\Lambda_{yx}$ (see Section 2) has unit Frobenius norm (this has no effect on optimization other than calibrating learning rate and standard deviation of initialization to their conventional ranges). With the training objective taking the form of Equation (1), we then computed $c$ — the global optimum — in accordance with the formula derived in Appendix A. In our experiments with linear neural networks, balanced initialization was implemented with the assignment written in step *(iii)* of Procedure 1. In the non-linear network experiment, we added, for each $j \in \{1, \ldots, N-1\}$, a random orthogonal matrix to the right of $W_j$, and its transpose to the left of $W_{j+1}$ — this assignment maintains the properties required from balanced initialization (see Footnote 7). During all experiments, whenever we applied grid search over learning rate, values between $10^{-4}$ and 1 (in regular logarithmic intervals) were tried.

