# OpenReview forum: "A Convergence Analysis of Gradient Descent for Deep Linear Neural Networks"
_ICLR.cc/2019/Conference_

### Official Review · AnonReviewer2 · 2018-10-26
**A Convergence Analysis of Gradient Descent for Deep Linear Neural Networks**

**Rating:** 7
**Confidence:** 4

**Review:**

Summary:

The paper provides the convergence analysis at linear rate of gradient descent to global minima for deep linear neural networks – the fully-connected neural networks with linear activation with l2 loss. The convergence only works under two necessary assumptions on initialization: “weight matrices at initialization are approximately balanced” and “the initial loss is smaller than the loss of any rank-deficient solution”. The result of this work is similar to that of Barlett et al. 2018, but the difference is that, in Barlett et al. 2018, they consider a subclass of linear neural networks (linear residual networks – a subclass of linear neural networks which the input, output and all hidden layers are the same dimensions).

Comments:

This paper focuses on theoretical aspect of Deep Learning. Yes, theoretical study of gradient-based optimization in deep learning is still open and needs to spread more. I have the following comments and questions to the author(s) and hope to discuss further during the rebuttal period:

1) Most of the deep learning applications are well-known used the neural networks with non-linear activation (specifically ReLU). Could you please provide any successful applications that linear neural networks could achieve better performance over the “non-linear” one? Yes, more layers may lead to better performance since we have more parameters. However, it is still not clear that which one is better between “linear” and “non-linear” with the same size of networks. I am not sure if this linear neural networks could generalize well.

2) For N=1, the problem should become linear regression with strongly convex loss, which means that there exists a unique W: y = W*x in order to minimize the loss. Hence, if W = W_N*....*W_1, the problem becomes non-convex w.r.t parameters W_N, ...., W_1 but all the minima could be global. Can you please provide some intuitions why the loss function could have saddle points? Also, is not easier to just solve the minimization problem on W?

3) Similar with l2 loss, it seems that the problem needs to be restricted on l2 loss. In understand that it could have in some applications. Do you try to think of different loss for example in binary classification problems?

4) I wonder about the constant “c > 0” in the definition 2 and it would use it to determine the learning rate. Do you think that in order to satisfy the definition 2 for the most cases, constant c would be (arbitrarily) small or may be very close to 0? If so, the convergence rate may be affected in this case.

5) The result of Theorem 2 is nice and seems new in term of probabilistic bound.  I did not see the similar result in the existing literature for neural networks.

6)  It would be nice if the author(s) could provide some experiments to verify the theory. I am also curious to know what performance it could achieve for this kind of networks.

I would love to discuss with the author(s) during the rebuttal period.

---

> ### Author Response · Authors · 2018-11-12
> **Response to reviewer 2**
>
> Thank you for the feedback.  Below are answers to your comments by their numbering.
>
> 1) Linear neural networks are of interest not for their practical importance (as stated in the introduction, they are no better than linear predictors), but because they are viewed as a first step in theoretical analysis of optimization for deep learning. Similar to non-linear neural networks, their loss landscape is highly non-convex with multiple minima and saddles. Hence the extensive past work on this model, e.g. [1]-[6] below.
>
> 2) The loss landscape of a linear neural network always includes saddle points (non-strict saddles if depth > 2). Example: network with all-zero weights, where gradient vanishes but (in any reasonable setting) there exist directions that will decrease the loss. As you suggest, it is indeed easier to optimize the end-to-end model directly (convex program), but the entire point in studying linear neural networks is analysis of gradient descent on the non-convex loss.
>
> 3) Our paper --- specifically the analysis of discrete updates --- treats only L2 loss (as do [1]-[4] below), but can be extended to cover any smooth and strongly convex loss. The more idealized analysis of gradient flow can allow non-strongly convex losses as well, in particular ones used for classification (e.g. cross-entropy). We will mention this in the text.
>
> 4) The constant $c$ in Definition 2 --- deficiency margin --- indeed affects our established convergence rate. The likelihood of it being sufficiently large under random initialization is a major topic in the paper, addressed extensively throughout Section 3 and Appendix B. In a nutshell, one can ensure significant deficiency margin (with constant probability), but that may come at the expense of approximate balancedness (Definition 1). The challenge is to satisfy both conditions, and for that we define the balanced initialization in Section 3.3.
>
> 6) We will add an appendix with some numerical experiments demonstrating our findings.  Note that in general, the performance of linear neural networks has been evaluated in various papers (e.g. [1] and [5] below).
>
>
> Finally, with regards to your summary of our paper, we would like to point out that linear residual networks --- the subclass of linear neural networks for which convergence has previously been established (by Bartlett et al.) --- are characterized not only by the input, output and hidden dimensions being the same, but also by a restriction to the specific initialization of identity.
>
>
> [1] Exact solutions to the nonlinear dynamics of learning in deep linear neural networks.  Saxe et al.  ICLR 2014.
> [2] Deep learning without poor local minima.  Kawaguchi.  NIPS 2016.
> [3] Identity matters in deep learning.  Hardt and Ma.  ICLR 2016.
> [4] Gradient descent with identity initialization efficiently learns positive definite linear transformations.  Bartlett et al.  ICML 2018.
> [5] On the optimization of deep networks: Implicit acceleration by overparameterization.  Arora et al.  ICML 2018.
> [6] Deep linear networks with arbitrary loss: All local minima are global.  Laurent and Brecht.  ICML 2018.

---

> > ### Comment · AnonReviewer2 · 2018-11-12
> > **Reply**
> >
> > Dear author(s),
> >
> > Thank you for your response!
> >
> > Yes, I do agree that theoretical aspects of neural network need to be investigated and spread more. I also agree that you have a nice theoretical result for deep linear neural network. My only concern is just whether this network could have some impact in practice and why we have to use it instead of doing minimization "end-to-end" model.
> >
> > Although there are still some limitations of the results, I hope this paper could influence others to expand more theoretical sides of neural networks rather than just experiments.
> >
> > I will likely increase my score for your paper!

---

> > > ### Author Response · Authors · 2018-11-12
> > > **Thank you**
> > >
> > > Thank you for the positive feedback! We plan to add an experimental section before deadline which will illustrate useful facts about the dynamics and support our theory. That may also be of interest.

---

> > > > ### Author Response · Authors · 2018-11-27
> > > > **Experiments added**
> > > >
> > > > See above

---

> > > ### Author Response · Authors · 2018-11-27
> > > **Experiments**
> > >
> > > We have added experiments to our submission which we believe may address your comment regarding impact on deep learning practice.  The procedure of balanced initialization, motivated by our theory, leads to improved (faster and more stable) convergence in the settings we evaluated.  We hope this will inspire similar ideas that will lead to improvements in larger, state-of-the-art settings.

---

> > > > ### Comment · AnonReviewer2 · 2018-11-27
> > > > **Response**
> > > >
> > > > Thank you, I have changed my score!

---

> > > > > ### Author Response · Authors · 2018-11-27
> > > > > **Thank you**
> > > > >
> > > > > Thank you for the swift response and positive feedback!

---

### Official Review · AnonReviewer1 · 2018-11-03
**Well-written paper, I would vote for acceptance, but I have some concerns as well.**

**Rating:** 7
**Confidence:** 5

**Review:**

This paper studies the convergence of gradient descent on the squared loss of deep linear neural networks. The authors prove linear convergence rate if (1) the network dimensions are big enough so that the full product can have full rank, (2) the singular values of each weight matrices are approximately the same, (3) the initialized point is “close enough” to the target.

First of all, this paper is well-written. It reads smoothly, effectively presents the key ideas and implications of the result, and properly answers to possible concerns that arise while reading. The improvement over the previous work ([Bartlett et al 18’]) is quite substantial.

Deep linear neural networks are important, and having a good understanding of linear neural networks can provide us useful insights for understanding the more complex ones, i.e., the nonlinear neural networks. In that regard, I really liked the discussion at the end of Section 3.1. My general opinion for this paper is acceptance, but I also have a number of concerns and questions.

My main concern about the study of GD on linear neural network is whether we really get any “benefit” or “acceleration” from depth, i.e., is GD on linear neural nets any faster than GD on linear models. It’s been shown that we get acceleration in some cases (e.g., $\ell_p$ regression when $p>2$ [Arora et al. 18’]), but some other results (e.g., [Shamir 18’] mentioned in Section 5) show that GD on linear neural nets (when weight matrices are all scalar) suffer exponential (in depth) increase in convergence time at near zero region, due to the vanishing gradient phenomenon. From my understanding, this paper circumvents this problem by assuming deficiency margin, because in the setting of [Shamir 18’], deficiency margin means that the initialized product ($W_{1:N}$) has the same sign as $\Phi$ and far enough from zero, so we don’t have to pass through the near-zero region.

Even with the deficiency margin assumption, the exponential dependence in depth can also be observed in this paper, if we use independent initialization of each weight matrices. In Claim 3, in order to get the probability 0.49 result, the margin $c$ must be very small (O(1/N^N)) as N goes to infinity, resulting in very small $\delta$ and $\eta$ in Theorem 1, and convergence time $T$ exploding in depth. On the other hand, if we fix $0 < c < 1$, then the probability of satisfying deficiency margin will be smaller and smaller as $N$ increases. Is this “blow-up in N” problem due to the fact that the loss is l2? Or am I making false claims? I would like to hear the authors’ opinion about this.

The paper proposes a balanced initialization scheme that doesn’t suffer exponential blow up (Procedure 1 and Theorem 2), but even with this, the learning rate must decay to zero in polynomial rate in N, also resulting in polynomial increase in convergence time as depth increases. Moreover, this type of initialization scheme (specifically tailored for linear neural networks) is not what people would do in practice; we normally would initialize each layer at random, and may suffer the problems discussed in the above paragraph. That is why I’d love to hear about the authors’ future work on layer-wise independent initialization, as noted in the conclusion section.

Below, I’ll list specific concerns/questions/comments.
* In my opinion, the statements about “necessity” of two key assumptions are too strong, because the authors only provide counterexamples of non-convergence. As [Theorem 3, Shamir 18’] shows (although in scalar case), even when the assumptions are not satisfied, a convergence rate $O(exp(N) * log(1/\epsilon))$ is possible. It will be an interesting future work to clearly delineate the boundary between convergence and non-convergence.

* In Thm 2 and Claim 3, what happens if dimension $d_0$ is smaller? What is the reason that you had to restrict it to high dimension? Is it due to high variance with few samples?

* In Thm 2, constants $d’_0$ and $a$ hide the dependence of the result on p, but I would suggest stating the dependence of those parameters on p, and also dependence on other parameters such as N.

* In Section 5, there is a statement “This negative result, a theoretical manifestation of the “vanishing gradient problem”, is circumvented by balanced initialization.” Can you elaborate more on that? If my understanding is correct, there is still $\sigma_min$ multiplier in Eq (9), which means that at near-zero regions, the gradient will still vanish.

I appreciate the authors for their efforts, especially on the heavy math in the proof of the main theorem. I would like to hear your comments and/or corrections (especially on my “T blowing up in N” claim) and discuss further.

---

> ### Author Response · Authors · 2018-11-12
> **Response to reviewer 1**
>
> Thank you for the support and the very thoughtful review!
>
> ----
>
> With regards to the “T blowing up in N” matter:
>
> Please note that our paper only provides *upper bounds* on the time it takes gradient descent to converge. These bounds indeed increase with depth, but we do not claim they are tight. It is possible that while our bound becomes worse, optimization actually accelerates.
>
> As you have noted, if we assume initialization admits a fixed deficiency margin $c > 0$, our main convergence result suffers from polynomial deterioration in depth (balancedness needs to be tighter, learning rate smaller, and number of iterations larger). This corresponds to the case of balanced initialization (Procedure 1). When layers are initialized independently around zero, the deficiency margin decreases exponentially with depth, further lengthening convergence time by an exponential factor. In this sense our results comply with [1], though we provide only upper bounds (positive results), without lower bounds (negative results) like theirs.
>
> Because we provide only upper bounds, there is no contention between our results and [2], which claims that added depth can sometimes accelerate convergence. Moreover, to the best of our knowledge, [2] does not consider asymptotic dependence on depth. It shows (through a mix of theory and experiments) that increasing depth from one layer to two or three can accelerate gradient descent under $l_p$ regression with $p > 2$, while pointing out that even in such settings, additional layers can cause a “vanishing gradient problem” (which may be alleviated by replacing near-zero initialization with identity initialization). As you suggest, an additional point reconciling our results with those of [2] is the fact that in the setting we treat --- $l_2$ regression --- they did not find any acceleration by depth.
>
>
> [1] Exponential Convergence Time of Gradient Descent for One-Dimensional Deep Linear Neural Networks.  Shamir.  2018.
> [2] On the Optimization of Deep Networks: Implicit Acceleration by Overparameterization.  Arora et al.  ICML 2018.
>
> ----
>
> Answers to specific questions and comments (by the order in which they appear):
>
> * Throughout the paper, whenever we mention the necessity of our key assumptions (approximate balancedness and deficiency margin), we clearly state that this means a violation of any one of them *can* cause divergence. This is indeed not a strong form of necessity. A clear separation between all settings that lead to convergence and those leading to divergence would be extremely valuable; this ambitious goal is left for future work.
>
> * In Claim 3, the requirement $d_0 \geq 20$ is purely technical, designed to allow a more compact presentation of our results. It is used after Equation (47) in the appendix, to simplify expressions there.
>
> * In Theorem 2, the constants $d’_0$ and $a$ are not presented explicitly because their dependence on $p$ is implicit.  Namely, as discussed at the end of the proof (Appendix D.3), any choice of $d’_0$ and $a$ that ensures Equation (38) is greater than $p$ suffices. We give there an example demonstrating that these constants need not be large ($d’_0 = a = 100$ suffice for $p = 0.25$). We will add this information as a footnote below Theorem 2. Thank you for raising the matter!
>
> * In Section 5, the statement of balanced initialization (Procedure 1) circumventing the “vanishing gradient problem” means precisely what you mentioned in the beginning of your review --- as opposed to layer-wise independent initialization, balanced initialization is not prone to producing extremely small end-to-end matrices when the depth is large. This means that it can escape the near-zero region, thereby ensuring that $\sigma_min$ multiplier in Equation (9) is sufficiently large at initialization (our analysis then shows it will remain that way throughout optimization).

---

### Official Review · AnonReviewer3 · 2018-11-04
**nice theoretical result about deep linear neural networks**

**Rating:** 7
**Confidence:** 4

**Review:**

This paper continues the recent line of study on the convergence behaviour of gradient descent for deep linear neural networks. For more than 2 layers, the optimization problem is nonconvex and it is known strict saddle points exist. The main contribution is a relaxation of the balancedness condition in previous work by Arora et al and a new deficiency margin condition, which together allowed the authors to prove that gradient descent will converge to an epsilon solution in at most O(log 1/epsilon) iterations (under reasonable assumptions on step size and other parameters). Examples on how to satisfy the two conditions are discussed. Overall, the obtained results appear to be a solid contribution beyond our current understanding of deep linear neural networks, and potentially may be helpful towards our understanding of deep nonlinear neural networks.

This paper is very well-written. The authors gave an elegant short proof for the gradient flow case and spent efforts in proving the discretized version as well. The discussion of related works seems to be appropriate and thorough.

One thing I would love to see more discussions about is the deficiency margin assumption. I know the authors provided some argument about its necessity in the appendix, but is it possible that under the deficiency margin assumption, the nonconvex optimization problem is really "trivial" hence the linear convergence of gradient descent? For instance, can one prove that on this level set there still could be some (strict) saddle-point? And what if Phi is rank-deficient?

In the paragraph proceeding Section 3.2, the authors mentioned that "overly small standard deviation will render high magnitude for the deficiency margin, and therefore fast convergence improbable." Is there a typo here? Do you mean  a smaller deficiency margin? If not, can you please provide more details?

Lastly, it would be great if the authors could complement the theoretical results with some numerical experiments, especially to test the initialization strategies in Section 3.3.

---

> ### Author Response · Authors · 2018-11-12
> **Response to reviewer 3**
>
> Thank you for the positive and enlightening feedback!  Below we address your comments/questions by the order in which they appear.
>
> * The assumption of deficiency margin at initialization does not trivialize the optimization problem. It induces a sublevel set without saddles, but nonetheless, this sublevel set is unbounded, and the landscape over it is non-convex and non-smooth. Indeed, we show in Appendix C that initialization with deficiency margin alone is not enough to ensure convergence --- without approximate balancedness, the non-smoothness can cause divergence. This is an excellent question that will be addressed in an updated version of the paper.
>
> * In the case where $\Phi$ is rank deficient the problem can be reduced to a subspace in which it has full rank, and if the end-to-end matrix is initialized within that subspace our analysis holds. The scenario of initialization outside the subspace is currently not covered. We regard its treatment as a direction for future work. We will mention this issue in the text; thank you for raising the question!
>
> * In the paragraph closing Section 3.1, the sentence you quote means that when the standard deviation of initialization is very small, the deficiency margin (if exists) will be small as well, thus the convergence rate we provide will be slow. This accords with the “vanishing gradient problem”, by which small (near-zero) initialization can significantly hinder convergence. We realize from your question that in its current form the sentence may be confusing. It will be rephrased.
>
> * We will add an appendix with some numerical experiments demonstrating our findings.

---

### Author Response · Authors · 2018-11-27
**Submission update**

Submission was updated in accordance with our responses to reviewers --- several clarifications were added, and a new experimental section demonstrates the potential of balanced initialization to improve convergence in practice, analogously to its theoretical benefits.

---

### Meta-Review · Area_Chair1 · 2018-12-17
**Solid advance in convergence analysis of gradient descent for deep linear networks**

**Confidence:** 5
**Recommendation:** Accept (Poster)

**Metareview:**

This is a well written paper that contributes a clear advance to the understanding of how gradient descent behaves when training deep linear models.  Reviewers were unanimously supportive.